# *AutoScale*: Combining Multi-Task Optimization with Linear Scalarization

## Abstract

Multi-task learning is favored due to its efficiency and potential transfer learning achieved by sharing networks across tasks. While a series of multi-task optimization algorithms (MTOs) have been proposed to solve MTL optimization challenges and enhance performance, recent research claims that simple linear scalarization, which sums per-task loss with a carefully searched weight set, is sufficient, casting doubt on the added value of more complex MTO algorithms. In this paper, we provide a novel perspective that linear scalarization and MTOs are closely related and can be combined to yield high performance and efficiency. We show, for the first time, that a well-performing linear scalarization exhibits specific characteristics of certain optimization metrics proposed by MTOs, such as high task gradient magnitude similarity and low condition number, via an extensive empirical study. We then propose *AutoScale*, an efficient pipeline that leverages these influential metrics to guide the search for optimal linear scalarization weights. *AutoScale* shows superior performance than prior MTOs and performs close to the searched weight performance consistently across different datasets.

## 1 Introduction

Multi-task learning (MTL) has gained significant attention in deep learning, due to its efficiency in using a single network to learn multiple tasks, with acceptable or comparable performance to single-task learning (Chen et al., 2018; Liu et al., 2024; Xin et al., 2022; Hu et al., 2024). Especially for recent large-scale end-to-end models, MTL has become a convenient and attractive choice for users and maintainers (Hu et al., 2023).

However, multi-task optimization (MTO) issues, such as *gradient conflict* and *gradient dominance*, have been a challenge in MTL, which can lead to impartial learning, where tasks interfere and compete for limited shared representation power (Chen et al., 2018; Liu et al., 2021b; Yu et al., 2020; Senushkin et al., 2023; Ban & Ji, 2024; Lin et al., 2024). Over the years, researchers have developed a series of multi-task optimization algorithms (MTOs) and proposed metrics to analyze and quantify the optimization issues, subsequently used to guide the training process. Common MTO metrics describe loss scale balance (Chennupati et al., 2019; Liu et al., 2021b), gradient magnitude (Chen et al., 2018; Sener & Koltun, 2018; Liu et al., 2021b) and angle (Yu et al., 2020) similarities, stability (Senushkin et al., 2023), and task convergence progress (Guo et al., 2018). While MTOs claim superior performance compared to unitary scalarization, which sums per-task losses with equal weights, they are often criticized for their large computational and memory overhead due to per-task gradient calculations (Xin et al., 2022; Kurin et al., 2022).

Recently, Xin et al. (2022); Kurin et al. (2022); Royer et al. (2024) surprisingly observe that linear scalarization, which sums up the per-task loss with a fixed weight set, performs comparably or even superior to MTOs when the task weights are carefully chosen. This finding is notable because linear scalarization is conceptually and operationally simple, and it requires just a single backpropagation during training, despite the high computation cost of weight search. There is hence an ongoing debate about whether complicated MTO algorithms are necessary or even help.

To answer the question, we propose a novel perspective that bridges MTOs and linear scalarization: certain metrics proposed by MTOs, designed to quantify optimization issues of multi-task training, are useful in guiding the search for optimal linear scalarization weights. This offers a more efficient

alternative to weight search methods, such as grid search. Specifically, through extensive experiments, we show a strong correlation between linear scalarization performance and the key MTO metrics during training, as depicted in Figure 3. Based on the insights, we propose *AutoScale*, a two-phase automatic pipeline, which calculates an optimal weight set by optimizing key MTO metrics using gradient and loss information collected during the first training stage, and applies this fixed weight set for linear scalarization in the remaining second stage. We demonstrate the effectiveness of *AutoScale* across multiple datasets, including CityScapes (Cordts et al., 2016), NYUv2 (Silberman et al., 2012), and Nuscenes (Caesar et al., 2020).

To summarize, besides presenting a comprehensive summary of various MTO algorithms and metrics, our primary contributions are as follows:

- We identify, for the first time according to our knowledge, the relationship between MTO metrics and optimal linear scalarization: a well-performing linear scalarization typically exhibits specific characteristics of certain MTO metrics, such as high gradient similarity among tasks and low condition number, which could serve as reliable indicators for determining the optimal weight set.

- We introduce *AutoScale*, an efficient two-phase pipeline combining both MTOs and linear scalarization. Our method estimates an optimal linear scalarization weight set by optimizing key MTO metrics. Compared with gradient manipulating MTOs, our design reduces training time significantly.

- We conduct extensive experiments to show that *AutoScale* outperforms prior MTO methods in most cases, and performs close to the optimal linear scalarization, without the need for grid search, across various datasets including a large-scale autonomous driving dataset.

Upon publication, our code will be available as open-source.

## 2 RELATED WORK

### 2.1 MULTI-TASK LEARNING: OVERVIEW

Research in multi-task learning (MTL), particularly within deep learning, has largely focused on three main directions: (1) MTL-specific architectures, (2) task grouping, and (3) Multi-Task Optimization algorithms (MTOs). MTL-specific architecture aims to improve performance by designing customized network structures for better handling multiple tasks (Misra et al., 2016; Dai et al., 2016; Long et al., 2017; Ye & Xu, 2023). Task grouping, on the other hand, explores the relationships among tasks and reduces negative transfer by grouping non-conflicting or minimally-conflicting tasks during training (Thrun & O'Sullivan, 1996; Zamir et al., 2018; Standley et al., 2020). Lastly, MTOs address the problem by designing optimal algorithms to manipulate and combine task-specific gradients to update network parameters during back-propagation (Chen et al., 2018; Senushkin et al., 2023; Liu et al., 2024). In this work, we focus on the last approach considering both MTOs and linear scalarization.

### 2.2 MULTI-TASK OPTIMIZATION

We categorize MTL training issues into five types: (1) gradient dominance, (2) gradient conflict, (3) imbalanced convergence speed, (4) imbalanced loss, and (5) instability.

**Gradient Dominance.** Variations in the scale of task-wise gradients on the shared parameters create impartial learning outcomes (Liu et al., 2021b), where the network converged primarily on tasks with higher gradient magnitudes (as shown in Figure 1a). To tackle this, GradNorm (Chen et al., 2018) dynamically adjusts the task weights to ensure the norm of each task's scaled gradient is balanced. While IMTL-G (Liu et al., 2021b) approaches this by finding an aggregated gradient with equal projections onto each task gradient.

**Gradient Conflict.** Conflicting gradients with opposing directions (as shown in Figure 1b) could cause negative transfers (Senushkin et al., 2023; Lee et al., 2018). CosReg (Suteu & Guo, 2019) proposes a regularization term based on squared cosine similarity between tasks, penalizing the network when conflicting gradients are generated. PCGrad (Yu et al., 2020), on the other hand, avoids

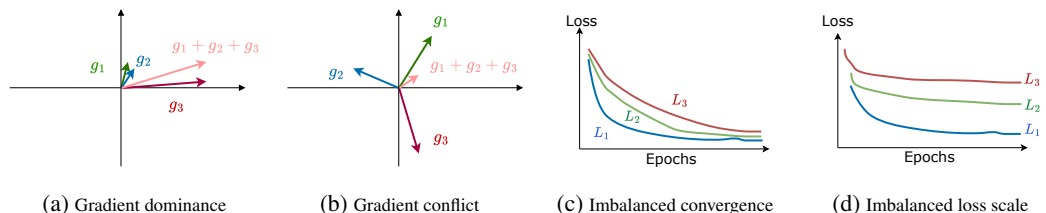

(a) Gradient dominance      (b) Gradient conflict      (c) Imbalanced convergence      (d) Imbalanced loss scale

Figure 1: Illustration of multi-task training issues. $g_i$ and $L_i$ represent gradient and loss for task $i$.

task conflicts by projecting the gradient of one task onto the normal plane of another. Similarly, Liu et al. (2021a) finds a conflict-averse direction to minimize overall conflicts, while GradDrop (Chen et al., 2020) enforces the sign consistency across task gradients to reduce conflict. Navon et al. (2022) tries to solve it as a Nash bargaining game.

**Imbalanced Convergence Speed.** Different tasks inherently have varying levels of difficulty, potentially leading to different convergence speeds (Guo et al., 2018; Yun & Cho, 2023) as shown in Figure 1c. To address this issue, methods like GradNorm (Chen et al., 2018), DTP (Guo et al., 2018), DWA (Liu et al., 2019), AMTL (Yun & Cho, 2023) and ExcessMTL (He et al., 2024) define specific measures of training convergence and adjust task weights based on these indicators. Additionally, Jacob et al. (2023) proposes to train single-task networks alongside the MTL network, using the convergence speed of the single-task network to guide online knowledge distillation.

**Imbalanced Loss.** Imbalances in the scale of task-specific losses (shown in Figure 1d) can result in suboptimal training outcomes. Many works have been focused on equalizing the scale of task losses. GLS (Chennupati et al., 2019) adopts geometric mean to prevent tasks with larger losses from dominating the overall loss. Following GLS, Yun & Cho (2023) proposes a weighted geometric mean of loss that is robust to scale variation. Liu et al. (2021b) proposes IMTL-L to derive task weights to balance re-scaled losses.

**Stability** Aligned-MTL (Senushkin et al., 2023) defines stability in MTL training as the stability of the linear system formed of task gradients. It proposes to stabilize the training process by aligning the principal components of the gradient matrix.

Additionally, we further discuss previously proposed MTO metrics to quantify and analyze these five MTL issues in section 3.1.

## 2.3 REVISITING LINEAR SCALARIZATION

In recent years, linear scalarization has been revisited and argued to be a superior alternative to more complex MTOs. Although linear scalarization has been shown to fail beyond the non-convex part of the Pareto front (Hu et al., 2024), studies such as Kurin et al. (2022); Xin et al. (2022); Elich et al. (2024); Royer et al. (2024) demonstrate that, in practice, it achieves performance comparable to or even better than other MTOs through large-scale experiments. However, a major open challenge for linear scalarization is identifying the optimal set of scalarization weights with minimal computational overhead. Although more efficient search methods have been proposed (Royer et al., 2024), they remain costly compared to directly applying existing MTOs due to requiring multiple training runs. In this work, we address this problem of costly search by proposing a unified pipeline to efficiently localize optimal scalarization weights with minimal overhead.

## 3 MTL METRICS IN LINEAR SCALARIZATION

Motivated by the ongoing debate between MTOs and linear scalarization in current literature, we conduct experiments (as shown in Tables 1 and 2) to compare the two approaches on multiple datasets. Our findings support the claim that linear scalarization performs as well as, if not better than, MTOs, as argued in Elich et al. (2024); Kurin et al. (2022); Xin et al. (2022). However, we acknowledge that weight search is challenging.

Existing works on MTOs, on the other hand, have made great efforts to reason and analyze potential issues in multi-task training, such as gradient conflicts, and have introduced various MTO metrics to

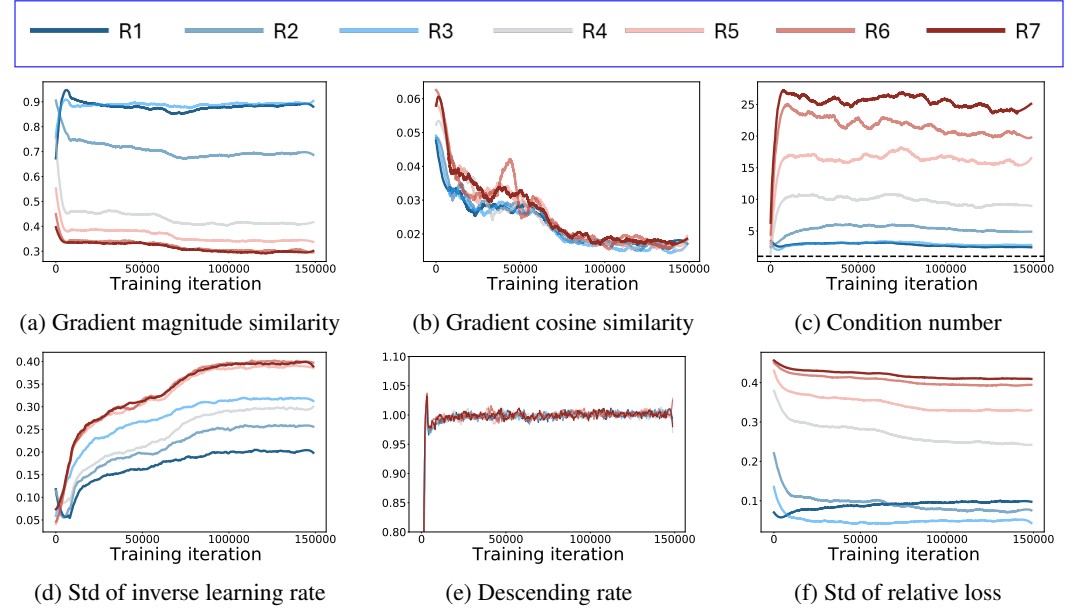

Figure 2: Evaluation on different MTO metrics and how they evolve during the training process of seven linear scalarization weight sets on the CityScapes dataset ~~: three with good performance (G), one moderate (M), and three with bad performance (B)~~ , with the performance ranking: ~~G1 > G2 > G3 > M > B3 > B2 > B1~~ R1 > R2 > R3 > R4 > R5 > R6 > R7. The metrics include (a) gradient dominance: gradient magnitude similarity; (b) gradient conflict: gradient cosine similarity; (c) training stability: condition number; (d,e) training progress: inverse learning rate, loss descending rate; (f) loss balance: relative loss scale. ⋆Unless specified, the metric values represent the average across tasks (or task pairs for metrics like similarity); captions with "std" indicate the standard deviation across tasks. The performance is ranked by $\Delta$m: measuring the average performance drop across tasks, as detailed in Section 5.

quantify the degree of these issues. In our work, we hypothesize that these metrics could be useful to guide the search for optimal linear scalarization weights.

As a first step, we summarize and categorize the metrics proposed by previous MTO studies in the following section.

## 3.1 MTL METRICS SUMMARY

**Gradient Dominance.** Gradient magnitude ratio $\frac{|g_1|}{|g_2|}$ between a pair of tasks has been used commonly to measure gradient dominance (Huang et al., 2023). Yu et al. (2020) proposes to quantify the gradient dominance of a pair of tasks via gradient magnitude similarity $\frac{2|g_1|\cdot|g_2|}{|g_1|^2+|g_2|^2} \in (0,1]$. A higher value indicates higher gradient magnitude similarity (thus less dominance), while a lower value reflects greater dominance.

**Gradient Conflict.** Previous works commonly define gradient conflict as when the cosine similarity between two task gradients is negative, $\frac{g_i \cdot g_j}{|g_i||g_j|} < 0$ (Suteu & Guo, 2019; Senushkin et al., 2023; Yu et al., 2020). In addition, Suteu & Guo (2019) proposes quantifying this issue via the standard deviation and mean over cosine similarities during training. Minimal conflict is indicated by both a low standard deviation and a mean close to zero. One could also measure gradient conflict by the cosine similarity between the task gradient and the update gradient (Liu et al., 2021b). We interpret negative values as the task receiving negative updates.

**Imbalanced Convergence Speed.** The convergence speed is defined in various ways. Grad-Norm (Chen et al., 2018) quantifies this via the inverse training rate, calculated as the ratio of the current training loss to the initial loss $l_t/l_0$. DWA (Liu et al., 2019) employs the ratio of losses between two consecutive epochs $l_t/l_{t-1}$, referred to as the loss descending rate. Similarly, FAMO (Liu et al., 2024) introduces the improvement ratio $(l_t - l_{t-1})/l_{t-1}$, which reflects the percentage change in training loss between successive epochs. Javaloy & Valera (2021) uses the rate of change in gradient magnitude to define task convergence speed. Guo et al. (2018) defines training progress

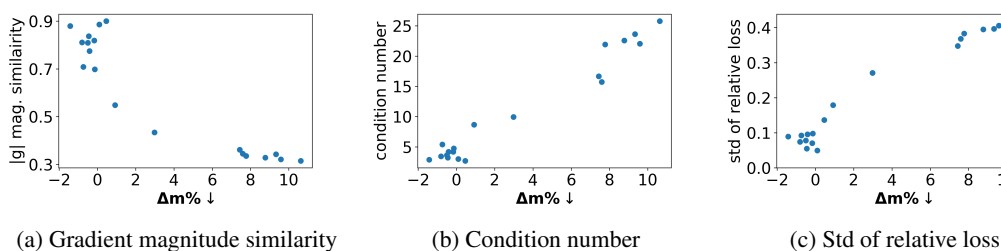

(a) Gradient magnitude similarity    (b) Condition number    (c) Std of relative loss

Figure 3: Performance ($\Delta\mathbf{m}$ ↓: average of performance drop compared to single-task learning, lower value indicate higher performance, as detailed in Section 5) vs. metrics values (average over training iterations) of 19 weight sets of linear scalarization. The plots illustrate a clear correlation between high performance and high gradient magnitude similarity, low condition number, and low standard deviation of relative loss among tasks.

using the notion of key performance indicator (KPI), in the range of $[0, 1]$, where values closer to 1 indicate higher progress. On the other hand, Yun & Cho (2023) views the performance of a single-task network as the optimal benchmark and uses the ratio between current multi-task performance and single-task performance to balance training. Likewise, Jacob et al. (2023) trains multi-task and single-task networks concurrently and use the ratio of their per-epoch performance as the convergence indicator.

**Loss Balance.** Different tasks in MTL can have loss terms with a wide range of scales. For example, the cross entropy loss applied for classification problems (Krizhevsky et al., 2012) typically falls under 1, whereas L2 loss for depth estimation (Zhang et al., 2023) could have much larger values, particularly when using millimeter units. One way to quantify loss balance is by the ratio between the losses of two tasks. Alternatively, one could define loss similarity by replacing the gradient magnitude $|gi|$ to loss values $l_i > 0$ in gradient magnitude similarity (Yu et al., 2020).

**Training Stability.** Senushkin et al. (2023) highlights the importance of training stability, which they measure using the condition number of the gradient matrix.

A complete list of metric summaries with mathematical formulas is provided in Appendix C.

### 3.2 SCALARIZATION WEIGHTS AND MTL METRICS

To investigate whether linear scalarization correlates with various metrics proposed by MTOs, we conduct extensive experiments using wide range of scalarization weights, observing the trajectories of MTL metrics during training as shown in Figure 2.

Surprisingly, we find that certain metrics, including gradient magnitude similarity, condition number, ~~inverse learning rate~~ and relative loss scale, serve as good indicators of performance, with clear patterns distinguishing ~~good~~high-performance from ~~bad~~low-performance linear scalarization sets. Specifically, the better-performing weight sets exhibit higher gradient magnitude similarity (Figure 2a). For training stability, the condition number of the best-performing linear scalarization is the lowest, approaching to 1 (Figure 2c). Regarding the training progress and loss balance, good weights lead to a more balanced convergence and loss scale across tasks, as reflected by smaller standard deviations in inverse learning rates (Figure 2d) and relative loss scales (Figure 2f). Figure 3 further illustrates the clear correlations between linear scalarization performance and key MTO metrics over 19 runs with different weight sets.

Conversely, the loss descending rate is less informative and could be discarded as a performance indicator (Figure 2e). Additionally, since linear scalarization only scales per-task loss, it does not affect the angle (cosine similarity) of the gradients between tasks (Figure 2b).

More MTO metrics visualization are provided in Appendix B.

## 4 METHOD

Given the strong relationship between linear scalarization performance and the key MTO metrics, where high performance corresponds to optimal MTO metric values, we hypothesize that the reverse also holds: linear scalarization with task weights that is expected to produce optimal MTO metrics values will lead to high performance. We can then leverage this to localize the optimal task weights by optimizing the key metric value. To formalize this, we express this in optimization terms as follows:

$$w^* = \arg\min_w \mathbb{E}[\mathbf{F}(w|\{\mathcal{G}\}, \{\mathcal{L}\})], \quad \text{s.t.} \sum_{i=1}^{K} w_i = K, \tag{1}$$

where $w = \begin{bmatrix} w_1 & w_2 & ... & w_K \end{bmatrix}^T$ is the vector of $K$ task weights for linear scalarization. $\{\mathcal{G}\}$ and $\{\mathcal{L}\}$ are the sets of task-wise gradients w.r.t. the shared model parameters, and task losses, collected over multiple training iterations. $\mathbf{F}(w)$ is a generalized cost function conditioning on data including task gradients and losses. It assigns lower values (rewards) to weights that produce MTO metric values positively correlated with high performance, and higher values (penalties) to those associated with negative performance, as suggested in Figure 3. For instance, a potential cost function could penalize weights that result in imbalanced magnitudes of scaled gradients across tasks. In Section 4.1, we define our proposed cost functions $\mathbf{F}(w)$ for three key MTO metrics.

---

**Algorithm 1** AutoScale.

---

**Require:** Existing MTOs (e.g., PcGrad, IMTL-G), total iterations $T$, exploration ratio $\alpha$, window size $\tau$, cost function $\mathbf{F}(w)$, weight predictor function $f$
  /* Phase 1: Exploration */
  Gradient set $\mathcal{G}$, Loss set $\mathcal{L} \leftarrow \emptyset$
  **for** $t \leftarrow 1 : \alpha T$ **do**
    Run MTOs($L$) $\rightarrow$ manipulate gradient $g$ / weight $w$
    $\mathcal{G} \leftarrow \mathcal{G} \cup \{g_t\}, \mathcal{L} \leftarrow \mathcal{L} \cup \{l_t\}$
  **end for**
  /* Phase 2: Linear Scalarization */
  Calculate weight for each sliding window of size $\tau$
  **for** $i = 1 : \alpha T/\tau$ **do**
    $w_i \leftarrow \arg\min_w \mathbf{F}(w|\{g_i : g_{i+\tau}\}, \{l_i : l_{i+\tau}\})$
  **end for**
  Determine fixed weight for rest $(1 - \alpha)T$ iterations
  $\hat{w}^* \leftarrow f(\{w_1, w_2, ..., w_{\alpha T/\tau}\})$
  **for** $t \leftarrow \alpha T + 1 : T$ **do**
    Run linear scalarization using $\hat{w}^*$
  **end for**

---

One could optimize Equation (1) over multiple training iterations, or across an entire training, or even by leveraging data from multiple runs to account for network randomness during training, to get a precise and robust optimal weight for a specific combination of dataset, tasks, and model. However, note that more iterations or runs mean increased computational costs.

We then propose *AutoScale*, an efficient and practical two-stage pipeline that partitions a single training run into two phases. The idea is to use the first phase's statistics to calculate an approximated optimal weight $\hat{w}^*$, which is then applied for linear scalarization in the remaining second phase. A summary of *AutoScale* is provided in Algorithm 1.

Specifically, in the first Eexploration phase, we run a selected MTO algorithm (e.g. PCGrad, IMTL-G) to collect training statistics, including gradients and losses, required for later weight optimization. We divide the Eexploration iterations into disjoint windows. For each window with index $i$, a local optimal weight set $w_i$ is calculated by optimizing key MTO metrics through minimizing the cost function $\mathbf{F}(w)$. Using the local weight sets calculated for each window in the Eexploration phase, we estimate an optimal weight set to be used in the subsequent Llinear Sscalarization phase. To do this, we apply a predictor, $f : \{w_i\} \mapsto \hat{w}^*$, which maps the derived local weight sets to a single output as the approximation of optimal weight. In Section 4.2, we introduce the specific design of the weight predictor $f(\{w_i\})$.

### 4.1 COST FUNCTIONS

We construct our cost function, which is by definition computed over $\tau$ iterations, as the average of per iteration cost function $\mathbf{F}^t(w)$ at iteration $t$: $\mathbf{F}(w) = \frac{1}{\tau} \sum_{t=i}^{i+\tau} \mathbf{F}^t(w)$. In this work, we propose and analyze three cost functions, based on the observations from the linear scalarization experiments, considering gradient magnitude similarity, loss similarity, and condition number.

**Gradient Magnitude Similarity Maximization (Equal $|G|$).** We define $\mathbf{F}^t(w) = |\mathbf{A^t}w|$, where $\mathbf{A}^t \in \mathrm{R}^{\binom{\mathrm{K}}{2} \times \mathrm{K}}$ contains the magnitudes of task gradients, with each row concerns a pair of tasks. Specifically,

$$\mathbf{A}^t_{\mathrm{row}(i,j),k} = \begin{cases} |g_i^t| & \text{if } k = i \\ -|g_j^t| & \text{if } k = j \\ 0 & \text{otherwise,} \end{cases} \quad \text{e.g. } \mathbf{A}^t|_{K=3} = \begin{bmatrix} |g_1^t| & -|g_2^t| & 0 \\ |g_1^t| & 0 & -|g_3^t| \\ 0 & |g_2^t| & -|g_3^t| \end{bmatrix} \quad (2)$$

in which $\mathrm{row}(i,j)$ refers to the row index assigned to the task pair $(i,j)$, $i \neq j$. $g_i^t$ is the gradient of task $i$ at iteration $t$. With this construction, the cost is minimized when a set of task weights results in equal magnitudes for all re-scaled task gradients.

**Loss Similarity Maximization (Equal $|l|$).** The objective is to find a set of weights that optimally balance the task loss scales. It follows the same formulation as Equation (2), with gradient magnitudes $|g_i^t|$ replaced by loss scales $|l_i^t|$.

**Condition Number Minimization (Low Cond.).** We defined the cost function as: $\mathbf{F}^t(w) = \kappa(\mathbf{G}_w^t) = \frac{\sigma_{max}}{\sigma_{min}}$, where $\kappa(\mathbf{X})$ denotes the condition number of a matrix $\mathbf{X}$, and $\mathbf{G}_w^t = [w_1 g_1^t \ w_2 g_2^t \ ... \ w_K g_K^t]$ is the gradient matrix consisting of scaled task gradients. Unlike Align-MTL (Senushkin et al., 2023), which manipulates both the direction and magnitude of gradients, we lower the condition number by rescaling the gradients using an optimal weight set.

## 4.2 WEIGHT PREDICTOR

In the second linear scalarization phase, as illustrated in Figure 4, we determine the weight for the rest $(1 - \alpha)T$ iterations. We base the decision on the locally optimized weights set $\{w\} = \{w_1, w_2, ..., w_{\alpha T/\tau}\}$ (marked as purple), calculated from the collected gradients and losses during the first $\alpha T$ iteration of exploration phase. We experiment with ~~five~~ four simple methods, represented by $f : \{w_i\} \mapsto \hat{w}^*$, ~~as illustrated in Figure 4~~. **1)** the average of all weight sets (Avg. W). $\frac{1}{\alpha T/\tau} \sum_{i=1}^{\alpha T/\tau} w_i$. **2)** the weight set from the last window (Last. W).

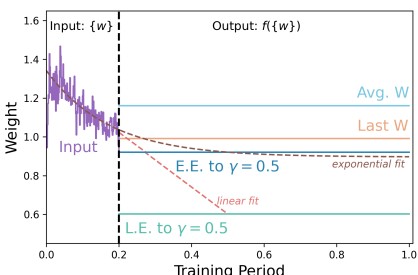

Figure 4: Illustration of various $f(\{w\})$.

$w_{\alpha T/\tau}$. **3)** the linearly extrapolated weight at iteration $\gamma T$ (L.E.). $f(\{w_i\}) = f_c^{\{w_i\}}(\gamma T)$, where $f_c^{\{w_i\}}(x) = ax + b$, is a line fitted using $\{w_i\}$. $\gamma$ represents the training progress ratio, $\gamma \in [0, 1]$, with $\gamma = 0$ at the start of training and $\gamma = 1$ at its completion. **4)** the exponentially extrapolated weight at iteration $\gamma T$ (E.E). Similar to 3), the linear equation is replaced with an exponential equation $f_c^{w_i}(x) = ae^{-bx} + c$, where $a$, $b$, and $c \in \mathbb{R}$ are the fitted curve parameters.

# 5 EXPERIMENT

We demonstrate the effectiveness of our proposed *AutoScale* compared with various baselines on different benchmarks.

**Datasets and Models.** We use three supervised MTL benchmarks, with a diverse range of dataset scales and number of tasks, to evaluate our proposed method: Nuscenes (2 tasks), CityScapes (3 tasks), and NYU-v2 (4 tasks). Nuscenes (Caesar et al., 2020) is a challenging large-scale outdoor benchmark for various autonomous driving tasks, among which we adopt 3D object detection and bird-eye-view (BEV) map segmentation. It contains more than 40k annotated multi-modality sample frames, each with six camera images and a 32-beam LiDAR pointcloud. We use UniTR (Wang et al., 2023) as the network architecture for our experiment. According to our knowledge, we are the first to systematically benchmark different MTOs in this scale of autonomous driving dataset, providing a wider cover of the study. CityScapes (Cordts et al., 2016) dataset contains 5k street-view RGB-D images with per-pixel annotations. We follow Senushkin et al. (2023) to use PSPNet (Zhao et al., 2017) on a three-task setup, namely disparity estimation, instance, and semantic segmentation. NYU-v2 (Silberman et al., 2012) is an indoor dataset consisting of 1449 RGB-D images and dense

Table 1: **Perception of traffic sence** (NUSCENES, two tasks, a large scale dataset). We report Unitr (Wang et al., 2023) model performance. Best scores are in gray, second-best in **bold**, and third-best underlined. [*]The performance reported for the searched weights represents the best result from 20 search trials. [†]s/iter denotes seconds per training iteration.

| Method | 3D Det.↑ mAP | NDS | Seg.↑ mIoU | MR ↓ | $\Delta\mathbf{m_{pos}}$ % ↓ | $\Delta\mathbf{m}$ % ↓ | Time s/iter[†] ↓ |
|---|---|---|---|---|---|---|---|
| STL Baseline | 0.693 | 0.725 | 0.701 | - | - | - | |
| *MTOs* | | | | | | | |
| UM | 0.681 | 0.716 | 0.698 | 5.5 | 1.91 | 0.95 | 0.455 |
| Gradnorm | 0.677 | 0.714 | 0.700 | 5.5 | 2.15 | 1.07 | 1.130 |
| MGDA | 0.647 | 0.696 | 0.660 | 10.0 | 11.15 | 5.57 | 1.157 |
| PCGrad | 0.671 | 0.711 | 0.657 | 9.5 | 8.82 | 4.41 | 1.170 |
| IMTL-G | 0.690 | 0.720 | 0.696 | 5.0 | 1.27 | 0.63 | 1.158 |
| RLW | 0.699 | 0.723 | 0.664 | 5.5 | 5.27 | 2.45 | 0.455 |
| Aligned-MTL | 0.664 | 0.706 | 0.680 | 8.5 | 6.50 | 3.25 | 1.213 |
| FAMO | 0.643 | 0.692 | 0.702 | 7.0 | 5.86 | 2.87 | 0.457 |
| *Linear Scalarization* | | | | | | | |
| Unitary | 0.699 | 0.729 | 0.680 | 4.0 | 2.98 | 1.14 | 0.453 |
| Searched weights* | 0.695 | 0.725 | 0.706 | 2.5 | 0.00 | -0.44 | |
| AutoScale (Ours) | 0.684 | 0.718 | 0.711 | **3.0** | **1.12** | -0.10 | 0.591 |

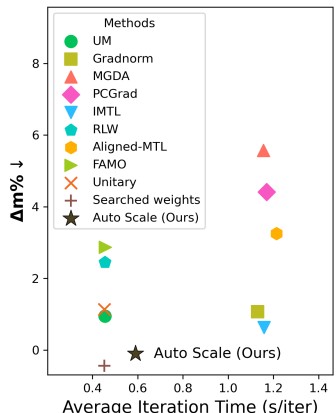

Figure 5: Performance ($\Delta\mathbf{m}$%) vs. training time for UniTR on Nuscenes. [*]The time for optimal weight search is **not** included and was obtained after **20** search trials.

per-pixel labeling with 13 classes. We adopt TaskPrompter (Ye & Xu, 2023), a state-of-the-art MTL model, and evaluate four scene understanding tasks: depth estimation, semantic segmentation, surface normal estimation, and edge prediction tasks. Further details of the experiment setup can be found in Appendix D.

**Baseline.** We compare our *AutoScale* with single-task learning (STL), UM (Kendall et al., 2018), GradNorm (Chen et al., 2018), MDGA (Sener & Koltun, 2018), IMTL-G (Liu et al., 2021b), PCGrad (Yu et al., 2020), RLW (Lin et al., 2021), Aligned-MTL (Senushkin et al., 2023), FAMO (Liu et al., 2024), unitary scalarization, and linear scalarization with the best set of task weights found by grid search.

**Evaluation Metrics.** Following previous methods (Senushkin et al., 2023; Liu et al., 2024), we use the Mean Rank (MR) and $\Delta\mathbf{m}$ metrics to evaluate multi-task performance. **1) $\Delta\mathbf{m}$ measures the average performance drop relative to the single-task baseline across all tasks.** $\Delta\mathbf{m} = \frac{1}{K}\sum_{k=1}^{K}(-1)^{\sigma_k}\delta\mathbf{m_k}$. Here, we denote $\delta\mathbf{m_k} = \frac{M_k - B_k}{B_k} \times 100$ as the performance difference on task $k$, where $M_k$ and $B_k$ are the $k$th task metric evaluated on a multi-task model and a single-task baseline respectively. $\sigma_k = 1$ if $M_k$ is higher the better, and $\sigma_k = 0$ otherwise. **2) Mean Rank (MR)** is the average ranking of performance across all tasks over all methods. For example, if a method ranks first on one task but second on the other task, MR $= (1 + 2)/2 = 1.5$.

In addition to the above conventional metrics, we propose a new metric $\Delta\mathbf{m_{pos}}$, which sums up all positive per-task performance changes $\delta m$, that is, total performance degradation: $\Delta\mathbf{m_{pos}} = \sum_{k=1}^{K}\max((-1)^{\sigma_k}\delta\mathbf{m_k}, 0)$. This metric captures the total percentage of performance drops $((-1)^{\sigma_k}\delta\mathbf{m_k} > 0)$ while disregarding improvements $((-1)^{\sigma_k}\delta\mathbf{m_k} < 0)$. When $\Delta\mathbf{m}$ is similar across methods, $\Delta\mathbf{m_{pos}}$ helps distinguish which methods minimize degradation, offering a lower bound on the percentage of tasks that perform worse. It offers valuable insight, particularly in scenarios where minimizing overall performance drops is prioritized over sacrificing some tasks' performance to enhance others.

**Our Implementation.** For all experiments on three benchmark datasets shown in Table 1 and Table 2, we use the following settings for our *AutoScale*: in the first exploration phase, we run IMTL-G (Liu et al., 2021b) to collect gradients and losses, with an exploration ratio $\alpha = 0.2$, window size $\tau = 50$, and a Gradient Magnitude Similarity Maximization constraint function $\mathbf{F}(\mathbf{w})$ (Equation 2). In the second linear scalarization phase, the weight function $f$ is a linear fit at training progress $\gamma = 0.5$.

**Results.** In most cases, the grid-searched linear scalarization weights yield the best performance across datasets in terms of MR, $\Delta\mathbf{m_{pos}}$%, and $\Delta\mathbf{m}$%. Our *AutoScale* achieves second-best performance on the large-scale Nuscenes dataset, outperforming all other MTOs and coming closest to the

Table 2: **Scene understanding**. CITYSCAPES: Three tasks with PSPNet (Zhao et al., 2017). NYUv2: Four tasks with TaskPrompter (Ye & Xu, 2023). Best scores are in gray , second-best in **bold**, and third-best underlined. *The performance reported for the searched weights represents the best result from 20 search trials, while for the others, it is the average of 3 random trials. [†]Gradnorm on NYUv2 produces negative weights, so we adjusted it to remain non-negative.

| Method | CITYSCAPES (three tasks) | | | | | | NYUv2 (four tasks) | | | | | | |
|---|---|---|---|---|---|---|---|---|---|---|---|---|---|
| | Sem. Seg. mIoU ↑ | Ins. Seg. L1 ↓ | Disp. MSE ↓ | MR ↓ | $\Delta \mathrm{m_{pos}}$ % ↓ | $\Delta \mathrm{m}$ % ↓ | Depth RMSE ↓ | Edge L1 ↓ | Normal Mean ↓ | Sem. Seg. mIoU ↑ | MR ↓ | $\Delta \mathrm{m_{pos}}$ % ↓ | $\Delta \mathrm{m}$ % ↓ |
| STL Baseline | 66.73 | 10.55 | 0.330 | - | - | - | 0.509 | 0.047 | 18.633 | 56.866 | - | - | - |
| *MTOs* | | | | | | | | | | | | | |
| UM | 57.96 | 9.99 | 0.361 | 5.33 | 22.41 | 5.69 | 0.497 | 0.048 | 19.325 | 56.892 | 4.50 | 4.61 | 0.55 |
| Gradnorm[†] | 52.53 | 10.06 | 0.395 | 8.33 | 40.99 | 12.11 | 0.513 | 0.047 | 18.971 | 55.583 | 5.75 | 5.04 | 1.26 |
| MGDA | 67.29 | 17.77 | 0.333 | 6.00 | 69.49 | 22.88 | 0.521 | 0.048 | 19.801 | 54.378 | 9.50 | 13.76 | 3.44 |
| PCGrad | 54.52 | 10.04 | 0.385 | 6.33 | 35.02 | 10.07 | 0.500 | 0.048 | 19.099 | 56.681 | 5.25 | 3.79 | 0.52 |
| IMTL-G | 65.44 | 10.70 | 0.326 | 6.33 | 3.37 | 0.71 | 0.498 | 0.048 | 19.224 | 56.222 | 7.25 | 5.41 | 0.83 |
| RLW | 52.69 | 10.12 | 0.405 | 8.67 | 43.83 | 13.27 | 0.499 | 0.048 | 19.380 | 56.504 | 8.00 | 6.15 | 1.02 |
| Aligned-MTL | 66.05 | 10.69 | 0.324 | **5.00** | 2.33 | 0.16 | 0.501 | 0.048 | 19.192 | 56.364 | 7.75 | 4.88 | 0.83 |
| FAMO | 66.02 | 10.25 | 0.327 | **5.00** | 1.07 | **-0.92** | 0.495 | 0.047 | 19.196 | 56.842 | 3.00 | 3.66 | **0.25** |
| *Linear Scalarization* | | | | | | | | | | | | | |
| Unitary | 54.16 | 9.96 | 0.392 | 6.33 | 37.47 | 10.62 | 0.499 | 0.048 | 19.150 | 56.765 | 5.75 | 4.12 | 0.56 |
| Searched weights* | 66.27 | 10.36 | 0.320 | 4.00 | 0.69 | -1.42 | 0.500 | 0.047 | 18.703 | 56.641 | **4.25** | 1.39 | -0.07 |
| AutoScale (Ours) | 66.31 | 10.58 | 0.328 | **5.00** | **0.93** | 0.10 | 0.501 | 0.047 | 19.104 | 56.733 | 5.00 | **3.42** | 0.45 |

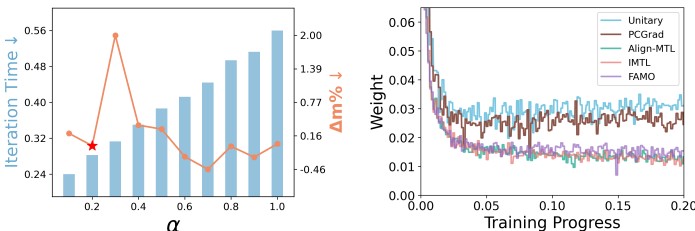

Figure 6: Ablation of using different values of the exploration ratio $\alpha$. The red star is our setting.

Figure 7: Weights computed with different MTOs in the 1st phase show notable differences.

Table 3: Ablation over different MTO algorithms selection in the first Exploration phase. IMTL-G shows good performance across metrics. Our default setting is marked in gray .

| MTOs | $\Delta \mathrm{m_{pos}}$% ↓ | $\Delta \mathrm{m}$% ↓ |
|---|---|---|
| Unitary | 11.52 | 2.63 |
| PCGrad | 7.73 | 2.53 |
| Align-MTL | 2.10 | 0.63 |
| IMTL-G | **0.93** | 0.10 |
| FAMO | 1.74 | **0.00** |

results of the searched weights in Table 1. For Cityscapes and NYUv2 datasets, as shown in Table 2, we achieved state-of-the-art results comparable to FAMO, trailing only the searched weights. In particular for the $\Delta \mathrm{m_{pos}}$%, *AutoScale* outputforms MTOs consistently.

**Efficiency.** Regarding training time, as shown in Figure 5 on large-scale dataset Nuscenes, gradient manipulating MTOs including GradNorm, MGDA, PCGrad, IMTL-G, and Aligned-MTL require three times the training time compared to linear scalarization. Since our *AutoScale* uses IMTL-G in the exploration phase with $\alpha = 0.2$, its training time is slightly longer than linear scalarization methods but it significantly reduce training time by over 45% compared with gradient manipulating MTOs, while delivering performance just behind the searched weights. The efficiency of *AutoScale* on the NYUv2 and CityScapes datasets is presented in Appendix D.1.

### 5.1 ABLATION STUDY

We conduct an extensive ablation of our *AutoScale* using the default setting outlined in Section 5. If not otherwise stated, the following experiments are based on CityScapes dataset (Cordts et al., 2016) with PSPnet (Zhao et al., 2017).

**Ablation on exploration ratio $\alpha$.** Figure 6 shows the impact of $\alpha$ on both performance and average training iteration time. A higher $\alpha$ results in the higher portion of training iterations being allocated to running MTOs and to collect loss and gradients for the exploration phase, which is computationally more demanding. We empirically find that an $\alpha$ of 0.2 strikes a good balance between computational time and performance. Though high $\alpha$ in general induces better performance than low $\alpha$, we argue that it would sacrifice the efficiency benefit of linear scalarization and therefore considered sub-optimal. Note that when $\alpha = 1$, it is equivalent to running the chosen MTO for the entire training.

**MTOs selection in exploration phase**. Since the selected MTO fascilitates the training of the network during early iterations in exploration phase, we argue that the choice of such MTOs is important that it does not drive the network to a poor local minimum. To illustrate this point, we perform ablation with five MTO methods: unitary scalarization, PCGrad (Yu et al., 2020), Aligned-MTL (Senushkin et al., 2023), IMTL-G (Liu et al., 2021b), and FAMO (Liu et al., 2024). The results, shown in Table 3 and Figure 7, reveal that different methods yield varying outcomes. The performance gap is clear: unitary scalarization and PCGrad perform noticeably worse compared to Aligned-MTL, IMTL-G, and FAMO, with IMTL-G and FAMO slightly outperforming Aligned-MTL. Aligned with the table results, Figure 7 also shows two distinct trends in the calculated weights based on the gradients collected during the exploration phase: unitary scalarization and PCGrad behave similarly, while the other three methods follow a different pattern. It highlights the importance of MTOs selection, as some methods are more prone to pitfalls such as converging to a suboptimal local minimum. Additionally, our findings suggest that certain MTO methods enhance our *AutoScale* pipeline's performance, offering evidence against earlier debates on the effectiveness of MTOs by helping avoid suboptimal solutions and improving optimization.

**Different constrain function $\mathbf{F}(\mathbf{w})$**. To calculate the optimized weight for the gradients and losses collected in the exploration phase, we experiment on different cost functions $\mathbf{F}(\mathbf{w})$, including optimizing for low condition number, equal loss scale $|L|$ and equal gradient magnitude $|g|$ among tasks. The results in Table 4 shows that using equal gradient magnitude gets a robust good performance over different datasets.

**Ablation on $f(\{w\})$**. We ablate five different weight predictors as introduced Section 4.2. Additionally, we test on the continuous linear fit until $\gamma = 0.5$ (L.E.$^\dagger$). As shown in Table 5, different datasets prefer different $f(w)$ methods. Overall, based on the mean rank (MR) across three datasets, the linear extrapolated value at a fixed point $\gamma$ shows the most robust and consistent performance.

**How metrics evolve during *AutoScale* training?** In Figure 2 and Appendix B.2, we show certain metrics evolve during different weight sets of linear scalarization. We also provide the key metrics trend during the training of *AutoScale* as shown in Appendix A. *AutoScale* exhibits favorable trends across different metrics, including a low condition number, balanced convergence speed (inverse learning rate), balanced loss scale, and equal angles to the final aggregated gradient, even when using the default cost function of equal gradient magnitude. It shows that these metrics are not independent, suggesting potential future work can be explored.

Table 4: Ablation over constrain function $\mathbf{F}(\mathbf{w})$. We show the performance of optimizing different metrics including low condition number, equal loss scale, and equal gradient magnitude over three datasets. Our default setting is marked in  gray .

| Cost Function | $\mathbf{\Delta m\%}\downarrow$ | | |
|---|---|---|---|
| | Nuscenes | NYUv2 | CityScapes |
| Low Cond. | 0.18 | 0.51 | 1.63 |
| Equal $|l|$ | 0.63 | 1.13 | **0.08** |
| Equal $|g|$ | **-0.10** | **0.45** | 0.10 |

Table 5: Ablation on $f(w)$ when $\gamma = 0.5$. $^\star$MR here refers to the average ranking of $\mathbf{\Delta m\%}$ across three datasets, not among different metrics. Our default setting is marked in  gray .

| $f(w)$ | $\mathbf{\Delta m\%}\downarrow$ | | | MR$^\star$ $\downarrow$ |
|---|---|---|---|---|
| | Nuscenes | NYUv2 | CityScapes | |
| Avg. W | 0.37 | 0.69 | -0.03 | 3.0 |
| Last W | 0.25 | 0.82 | -0.10 | 2.7 |
| L.E. | -0.10 | 0.45 | 0.10 | **2.3** |
| L.E.$^\dagger$ | 0.43 | 0.24 | 0.89 | 3.3 |
| E.E. | 0.95 | 0.36 | 0.26 | 3.7 |

## 6 CONCLUSION

In this work, we propose a novel perspective on the ongoing debate between MTO algorithms and linear scalarization. Through a comprehensive set of experiments, we identify that well-performing linear scalarization aligns with specific characteristics of certain MTO metrics, including high gradient magnitude similarity, low condition number, and more balanced loss scale across tasks. These findings help bridge the gap between linear scalarization and existing MTOs, highlighting the importance of both in addressing MTL training challenges. Building on the insights, we introduce *AutoScale*, an efficient pipeline which combines both: determine the optimal linear scalarization weights using MTL metrics in a two-phase way. *AutoScale* achieves state-of-the-art performance across a wide range of benchmarks including a large-scale modern autonomous driving dataset, trailing only the searched weights, but without the need of grid search.

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

## A HOW METRICS EVOLVE DURING *AutoScale* TRAINING?

We add the key metrics trajectories during the training of our *AutoScale*, based on Figure 2. As shown in below figure, *AutoScale* exhibits favorable trends across different metrics, including a low condition number, balanced convergence speed (inverse learning rate), balanced loss scale, and equal angles to the final aggregated gradient, even when using the default cost function of equal gradient magnitude. It is evident that these metrics are not independent, suggesting potential future work can be explored.

Additionally, we observe an interesting pattern with IMTL-G. When IMTL-G is used during the first 20% of the exploration phase ($\alpha = 0.2$), it achieves near-perfect gradient magnitude similarity (close to 1) and gradient cosine similarity with the final aggregated gradient (with low standard deviation among tasks). This aligns with IMTL-G's objective of enforcing equal gradient magnitudes and angle with the aggregated gradient. However, it sacrifices loss scale balance, as indicated by a high standard deviation in the relative loss scale among tasks during this phase.

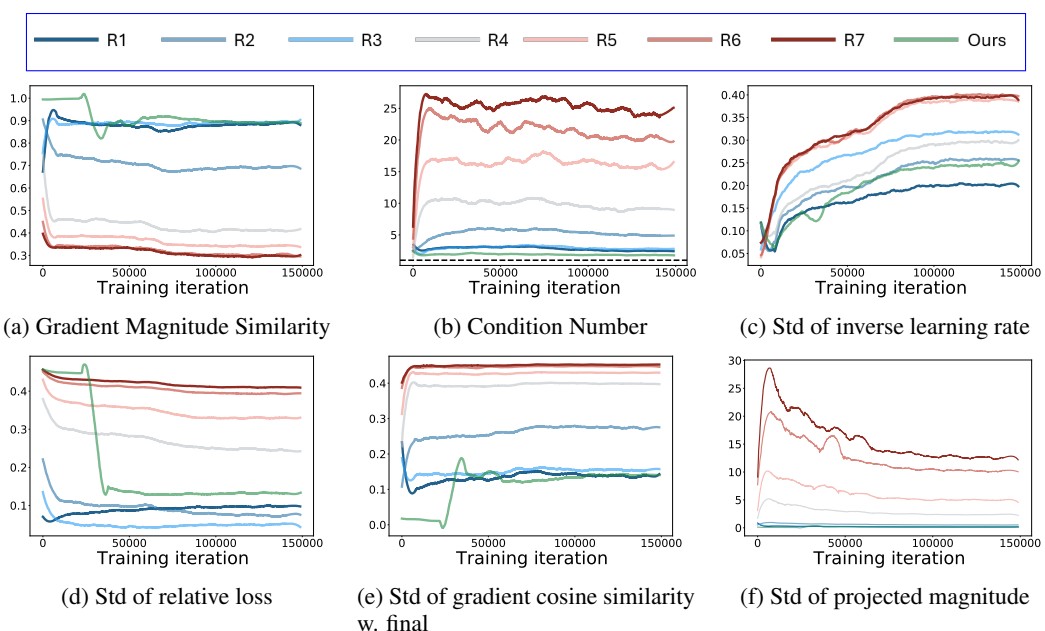

Figure 8: How metrics evolve during *AutoScale* training? In addition to the seven linear scalarization weight sets on the CityScapes dataset in Figure 2 ~~: three with good performance (G), one moderate (M), and three with bad performance (B)~~ , we include our *AutoScale* to observe how metrics behave. The performance ranking of all runs based on $\Delta m$ is: ~~R1 > G2 > ours > G3 > M > B3 > B2 > B1~~ R1 > R2 > ours > R3 > R4 > R5 > R6 > R7. *AutoScale* exhibits favorable trends across different metrics.

## B MORE METRICS VISUALIZATION IN LINEAR SCALARIZATION ACROSS VARIOUS DATASETS

### B.1 CITYSCAPES

In addition to the metrics presented in Figure 2 in CityScapes, track patterns of other metrics across multiple linear scalarizations runs with different task weights, as shown below.

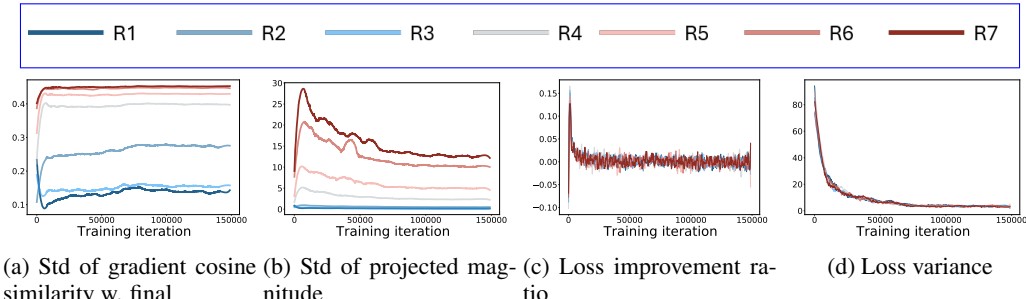

(a) Std of gradient cosine similarity w. final (b) Std of projected magnitude (c) Loss improvement ratio (d) Loss variance

Figure 9: Evaluation on different metrics and how they evolve during the training process of seven linear scalarization weight sets on the CityScapes dataset ~~: three with good performance (G), one moderate (S), and three with bad performance (B)~~ , with the performance ranking: ~~G1 > G2 > G3 > M > B3 > B2 > B1~~ R1 > R2 > R3 > R4 > R5 > R6 > R7. The metrics shown include: (a) cosine similarity between per-task gradient and final gradient (aggregated gradient from the weighted sum loss), (b) projected gradient magnitude of final gradient onto per-task gradient direction; (c) improvement ratio; and (d) loss scale variance. The figures illustrate how these metrics evolve during the training process on the CityScapes dataset. It is evident that cosine similarity with final and projected magnitude correlates with the performance of linear scalarization, whereas the loss improvement ratio and variance do not show such correlations.

## B.2 NUSCENES AND NYUv2

We present the behavior of various MTO metrics during linear scalarization on the Nuscenes and NYUv2 datasets below, similar to Figure 2 and Figure 3 on the CityScapes dataset.

Certain MTO metrics, including gradient magnitude similarity and condition number, consistently show strong correlations with the performance across both datasets. Poor-performing linear scalarization runs are always associated with highly unbalanced loss scales. In contrast, metrics such as loss variance and gradient cosine similarity (as linear scalarization does not alter per-task gradient directions) consistently show no correlation with performance.

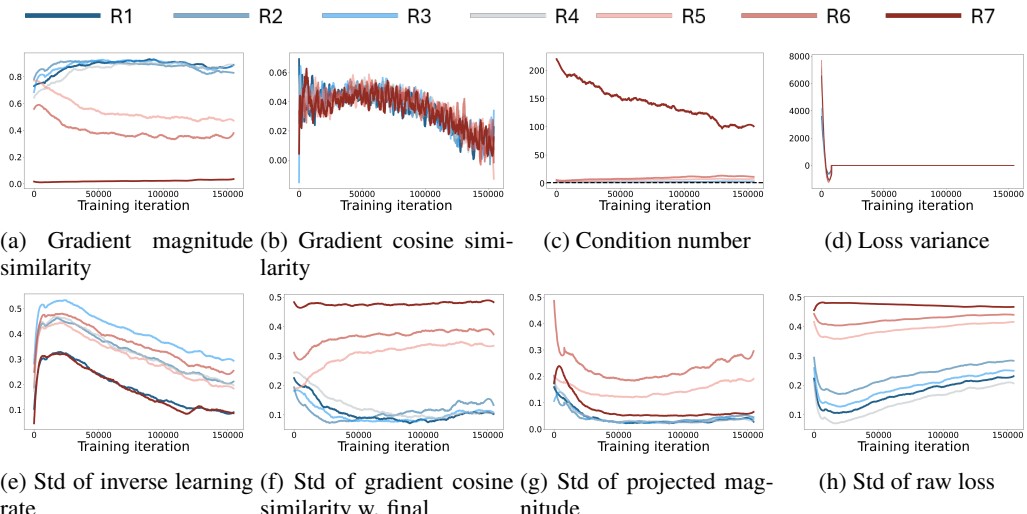

(a) Gradient magnitude similarity (b) Gradient cosine similarity (c) Condition number (d) Loss variance

(e) Std of inverse learning rate (f) Std of gradient cosine similarity w. final (g) Std of projected magnitude (h) Std of raw loss

Figure 10: Evaluation on different metrics and how they evolve during the training process of seven linear scalarization weight sets on the Nuscenes dataset, with the performance ranking: R1 > R2 > R3 > R4 > R5 > R6 > R7.

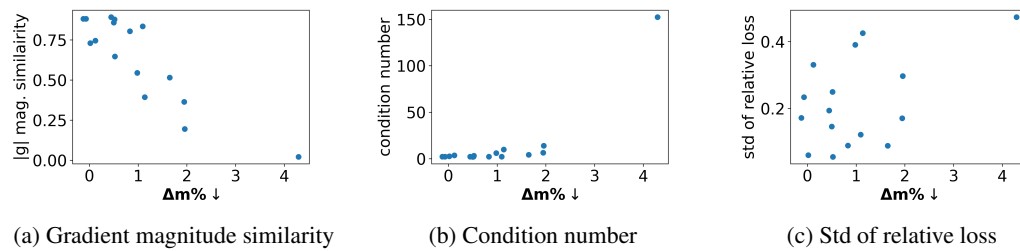

(a) Gradient magnitude similarity     (b) Condition number     (c) Std of relative loss

Figure 11: Nuscenes: Performance ($\mathbf{\Delta m} \downarrow$) vs. metrics values (average over training iterations) of 16 weight sets of linear scalarization.

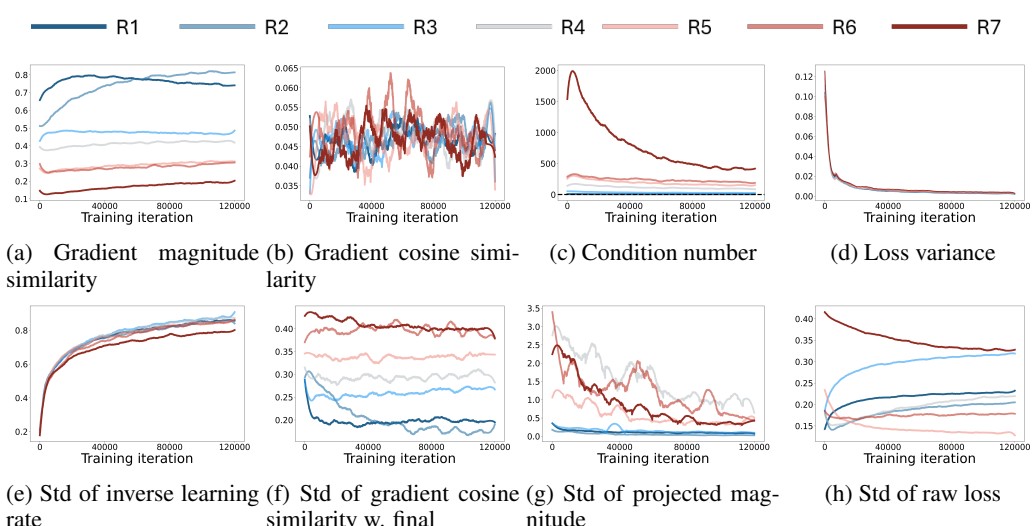

(a) Gradient magnitude similarity   (b) Gradient cosine similarity   (c) Condition number   (d) Loss variance

(e) Std of inverse learning rate   (f) Std of gradient cosine similarity w. final   (g) Std of projected magnitude   (h) Std of raw loss

Figure 12: Evaluation on different metrics and how they evolve during the training process of seven linear scalarization weight sets on the NYUv2 dataset, with the performance ranking: R1 > R2 > R3 > R4 > R5 > R6 > R7.

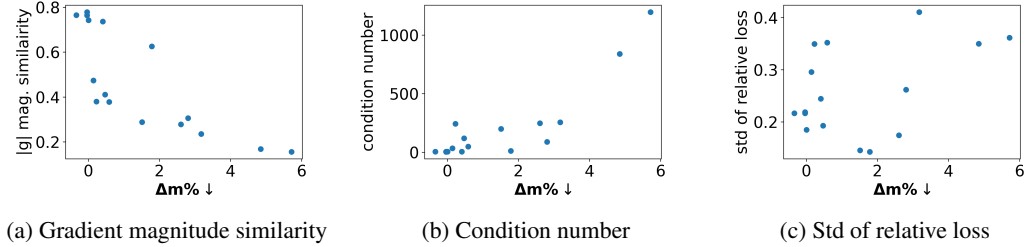

(a) Gradient magnitude similarity     (b) Condition number     (c) Std of relative loss

Figure 13: NYUv2: Performance ($\mathbf{\Delta m} \downarrow$) vs. metrics values (average over training iterations) of 16 weight sets of linear scalarization.

## C   LIST OF METRICS

We list the various metrics to quantify the degree of categorized multi-task training issues, with their mathematical formulas below.

Note that we will omit the iteration index $t$ whenever we use all items from the same iteration.

| Symbol | Description |
|---|---|
| $l_i(t)$ | The loss of task $i$ at time $t >= 0$. |
| $g_i(t)$ | The gradient of $l_i(t)$ w.r.t. the shared parameters $\theta_{\text{shared}}$. |
| $K$ | The total number of tasks. |
| $\|x\|$ | The Euclidean norm of a vector $x$. |
| $\theta_{i,j}$ | The angle (in radians) between two task gradient vectors $g_i$ and $g_j$. |

Table 6: **Notations**

## C.1 GRADIENT DOMINANCE

**Gradient Magnitude Ratio ($\gamma_{i,j}$) (Huang et al., 2023)**

$$\gamma_{i,j} = \frac{|g_i|}{|g_j|}, \text{ s.t. } |g_i| <= |g_j|$$

**Gradient Magnitude Similarity ($\Phi(g_i, g_j)$) (Yu et al., 2020)**

$$\Phi(g_i, g_j) = \frac{2|g_i||g_j|}{|g_i|^2 + |g_j|^2}$$

## C.2 GRADIENT CONFLICT

**Cosine Similarity to Average Gradient Direction ($\cos(\theta_i)$) (Javaloy & Valera, 2021)**

$$\cos(\bar{\theta}_i) = (\frac{g_i^T \bar{g}}{|g_i||\bar{g}|}),$$

where $\bar{g} = \frac{1}{K} \sum_{j=1}^{K} g_j$

**Cosine Similarity ($\cos(\theta_{i,j})$) (Yu et al., 2020)**

$$\cos(\theta_{i,j}) = \frac{g_i^T g_j}{|g_i||g_j|}$$

**Cosine Similarity to Final Gradient ($\hat{\cos}(\theta_i)$) (Liu et al., 2021b)**

$$\hat{\cos}(\hat{\theta}_i) = (\frac{g_i^T \bar{g}_0}{|g_i||\bar{g}_0|}),$$

where $\bar{g}_0$ is the final gradient used to update the shared network parameters, for example, under linear scalarization, $\bar{g}_0 = \sum_{j=1}^{K} w_j g_j$

## C.3 IMBALANCED CONVERGENCE SPEED

**Inverse Training Rate ($r_i(t)$) (Chen et al., 2018)**

$$r_i(t) = \frac{l_i(t)}{l_i(0)}$$

**Loss Descending Rate ($\eta_i(t)$) (Liu et al., 2019)**

$$\eta_i(t) = \frac{l_i(t)}{l_i(t-1)}$$

Note that in our implementation, to obtain a more meaningful and stable trajectory, we use the losses computed over a window of size $\tau$, that is:

$$\eta_i^\dagger(t) = \frac{\hat{l}_i(t)}{\hat{l}_i(t-1)},$$

where

$$\hat{l}_i(t) = \frac{1}{\tau} \sum_{n=t}^{t+\tau} l_i(n).$$

**Improvement Ratio ($\bar{r}_i(t)$) (Liu et al., 2024)**

$$\bar{r}_i(t) = \frac{l_i(t) - l_i(t+1)}{l_i(t)}$$

Note that in our implementation, similar to the Loss Descending Rate, we use loss over a window for the stability of the metric values:

$$\bar{r}_i^\dagger(t) = \frac{\hat{l}_i(t) - \hat{l}_i(t+1)}{\hat{l}_i(t)}$$

**Relative Inverse Training Rate ($\tilde{r}_i$) (Chen et al., 2018)**

$$\tilde{r}_i = \frac{K \cdot r_i}{\sum_{j=1}^{K} r_j}$$

Note that using this idea, we can compute any normalized (i.e. relative) task-wise metrics in the following general form:

$$\tilde{\beta}_i = \frac{K \cdot \beta_i}{\sum_{j=1}^{K} \beta_j},$$

where $\beta_i$ is some metric computed for task $i$.

**Task Loss Variance ($\sigma_i^2(t)$) (Kumar et al., 2021)**

$$\sigma_i^2(t) = \frac{1}{\tau - 1} \sum_{k=0}^{\tau-1} (l_i(t-k) - \bar{l}_i(t))^2,$$

where $\bar{l}_i(t)$ is the mean loss within the window:

$$\bar{l}_i(t) = \sum_{k=0}^{\tau-1} l_i(t-k)/\tau,$$

and $\tau$ is the window size.

**Focal Loss ($\text{FL}(\bar{k}_i, \alpha_i)$) (Guo et al., 2018)**

$$\text{FL}(\bar{k}_i, \alpha_i) = -(1 - \bar{k}_i^{\alpha_i}) \cdot \log(\bar{k}_i)$$

where $\bar{k}_i$ is the KPI of task $i$, defined to be within the range of $(0, 1)$, higher value should indicate better performance at time $t$. $\alpha_i$ is the focusing factor for task $i$, which adjusts the rate at which easy (good performance) tasks are down-weighted.

**Achievement ($a_i$) (Yun & Cho, 2023)**

$$a_i = (1 - \frac{Acc_i}{m \cdot p_i})^\gamma$$

, where $p_i$ is the potential of task $i$, usually defined as the single task accuracy. $m$ defines a safety margin considering the multi-task performance can potentially become larger than that of the potential. $\gamma$ is the focusing factor as in the focal loss.

**Training Progress ($m_i$) (Jacob et al., 2023)**

$$m_i = \frac{l_i^{\text{MTL}}}{l_i^{\text{STL}}}$$

**Relative Training Progress ($\lambda_i$) (Jacob et al., 2023)**

$$\lambda_i = K \frac{\exp(m_i/\tau)}{\sum_{j=1}^{K} \exp((m_j/\tau))}$$

**Relative Gradient Magnitude ($\bar{g}_i(t)$) (Javaloy & Valera, 2021)**

$$\tilde{g}_i(t) = \frac{|g_i(t)|}{|g_i(0)|}$$

### C.4  LOSS SCALE BALANCE

**Relative Loss Scale ($\tilde{l}_i$)**

$$\tilde{l}_i = \frac{l_i}{\sum_{j=1}^{T} l_j}$$

or

$$\tilde{l}_i = \frac{\exp\{l_i\}}{\sum_{j=1}^{T} \exp\{l_j\}}$$

**Loss Ratio ($r^l_{(i,j)}$)**

$$r^l_{(i,j)} = \frac{l_i}{l_j}$$

### C.5  TRAINING STABILITY

**Condition Number ($k(\mathbf{G})$) (Senushkin et al., 2023)**

$$k(\mathbf{G}) = \frac{\sigma_{\max}}{\sigma_{\min}},$$

where $\sigma$ are the singular values of the gradient matrix $\mathbf{G}$.

## D  EXPERIMENT DETAILS

**Nuscenes** For UniTR (Wang et al., 2023), while the model is designed to support both 3D detection and map segmentation, these tasks are not trained jointly. The reported results are based on single-task training, each optimized with different hyperparameters, such as varying epochs (10 for detection, 20 for segmentation), learning rates (3e-3 vs. 1e-3), and distinct data augmentations for detection and segmentation. To ensure that all of the experiments are conducted under the same training conditions, we apply the original detection configuration to both tasks: 10 epochs with a learning rate of 3e-3. Note that with this setup, we observe a performance drop in map segmentation compared to the original UniTR results, with mIoU decreasing from 0.732 to 0.701. We modify the network to include both task heads and train them simultaneously using the same configuration for the multi-task learning experiments. All experiments are done with $8 \times$ A100 GPUs.

**CityScapes** We adopt the same experiment setup as in Senushkin et al. (2023). The PSPNet (Zhao et al., 2017) is trained for 100 epochs with a learning rate of 1e-4 and a batch size of 8 on a single A100 GPU.

**NYUv2** We adopt TaskPrompter (Ye & Xu, 2023) for our experiments on NYUv2 dataset. The network is trained for 40000 iterations, with a learning rate of 1e-3, polynomial learning rate scheduling with weight decay of 1e-6, and a batch size of 2 on a single A100 GPU.

### D.1  RUNTIME

We present the runtime table across various datasets below. As *AutoScale* has in two phases—running an existing MTO in the exploration phase and using pure linear scalarization in the second phase—its runtime varies depending on the selected MTO. Generally, *AutoScale* is more efficient than gradient manipulating MTO algorithms such as GradNorm, MGDA, IMTL-G, and Aligned-MTL, which require gradient computation throughout the entire training process.

| Method | Nuscenes† | | CityScapes | | NYUv2 | |
|---|---|---|---|---|---|---|
| | Iter. Time (s) | Relative Time | Iter. Time (s) | Relative Time | Iter. Time (s) | Relative Time |
| *Linear Scalization* | | | | | | |
| Unitary | 0.453 | 1.00 | 0.195 | 1.00 | 0.298 | 1.00 |
| Searched weights‡ | | | | | | |
| *MTOs* | | | | | | |
| UM | 0.455 | 1.01 | 0.199 | 1.02 | 0.367 | 1.23 |
| Gradnorm | 1.130 | 2.50 | 0.572 | 2.93 | 0.790 | 2.65 |
| MGDA | 1.157 | 2.56 | 0.446 | 2.29 | 0.747 | 2.51 |
| PCGrad | 1.170 | 2.59 | 0.416 | 2.13 | 0.765 | 2.57 |
| IMTL | 1.158 | 2.56 | 0.422 | 2.16 | 0.829 | 2.78 |
| RLW | 0.455 | 1.01 | 0.190 | 0.97 | 0.287 | 0.96 |
| Aligned-MTL | 1.213 | 2.68 | 0.430 | 2.21 | 4.144 | 13.91 |
| FAMO | 0.457 | 1.01 | 0.198 | 1.02 | 0.290 | 0.97 |
| AutoScale⋆ (Ours) | 0.591 | 1.31 | 0.261 | 1.34 | 0.431 | 1.45 |

Table 7: Runtime comparison. ⋆ The runtime for AutoScale depends on the choice of MTO algorithm in the exploration phase. By default, it uses IMTL-G, resulting in a total runtime of approximately 20% of IMTL-G's time plus 80% of linear scalarization's time. † The runtime for Nuscenes is measured on 8 GPUs, while the others use a single GPU. ‡ For the searched weights, the runtime increases when the number of search trials increases. "Iter. Time" refers to the training iteration time.

