# OpenReview forum: "AutoScale: Combining Multi-Task Optimization with Linear Scalarization"
_ICLR.cc/2025/Conference — Submitted to ICLR 2025_

### Official Review · Reviewer_CV7C · 2024-11-01

**Soundness:** 3
**Presentation:** 3
**Contribution:** 2
**Rating:** 6
**Confidence:** 2

**Summary:**

The paper proposes AutoScale, a new approach that combines MTO with linear scalarization for efficient and effective MTL. The authors explore the synergy between MTOs and linear scalarization, proposing that certain MTO metrics can guide weight selection to improve scalarization efficiency. AutoScale operates in two stages: an exploration phase to gather gradient and loss data using an MTO method, and a scalarization phase that leverages optimized weights derived from key MTO metrics. Extensive experiments demonstrate that AutoScale outperforms prior MTO methods and nearly matches the performance of grid-searched scalarization weights, without the associated search costs.

**Strengths:**

The paper is well-organized and the idea is clear.
The presentation is overall clear, and the experiments seem extensive and convincing.

**Weaknesses:**

1. The technical contributions seem limited since the proposed methodology is pretty straightforward.
2. In the experiments, the number of tasks are too small (i.e. up to 4 tasks), which are not common in multi-task learning settings.
3. The code is unavailable, and it is impossible to reproduce the results.
4. Some typos. For example, Line 306 Exploration -> exploration
Line 311: Linear Scalarization -> linear scalarization

**Questions:**

1. Can authors discuss why linear scalaization correlates with some MTO metrics while is independent of others. What are commonality of these correlated MTO metrics?

2. What are benefits of Autoscale compared with traditional MTO algorithms?

---

> ### Author Response · Authors · 2024-11-23
> **Response to Reviewer CV7C - W1~W4**
>
> Thank you for your thoughtful review and for recognizing the strengths of our work, including its organization, clarity, and extensive experiments. We greatly appreciate your positive feedback and the time you have taken. Below, we address your concerns and questions in detail.
>
> $\ $
>
> **W1 Technical contribution:**
>
> We highlight our contributions in the common reply.  While we agree that our methodology is straightforward, we view this simplicity as a strength rather than a weakness.
>
> To the best of our knowledge, **no systematic analysis of MTO algorithms metrics' behavior during linear scalarization has been conducted before**.
> Through extensive experiments, we identify strong correlations between specific MTO metrics and linear scalarization performance.
> Inspired by the analysis, we propose *AutoScale*, marking the first instance of using different MTO metrics to determine linear scalarization weights.
>
> While the concept is simple, it addresses an important gap in the literature and offers a practical, effective solution for multi-task learning.
> Furthermore, our framework introduces a novel perspective, encouraging future exploration of the potential to leverage the core principles of various MTO algorithms to assist in weight searching for linear scalarization.
>
> $\ $
>
> **W2 Number of tasks:**
>
> Many commonly used multi-task learning benchmarks consist 2-4 tasks (e.g., CityScapes and NYUv2 as we show, as well as other popular datasets like Office-31 with 3 tasks and Office-Home dataset with 4 tasks), and these are frequently used in multi-task learning papers, such as Aligned-MTL [1]. Nevertheless, recent works indeed often include benchmarks with a higher number of tasks, such as QM9 dataset [2], having 11 tasks. Therefore, we conducted additional experiments on QM9 dataset, shown in the below table. We adopt the code base published by the authors of FAMO [3].
> Note that we excluded IMTL and MGDA results due to unresolved issues in the code base within the limited rebuttal period.
> For the performances of IMTL and MGDA on QM9, we suggest referring to the original FAMO.
>
> Moreover, given the limited time, we ran our method directly using the default hyper-parameters (with IMTL changed to FAMO only, due to the code base issue on IMTL), perhaps further tuning would improve the performance of our method more. Despite this, our *AutoScale* ranks second among all the baselines presented.
>
>
>
>
> | Method | $\mu$| $\alpha$   | $\epsilon_{\text{HOMO}}$ | $\epsilon_{\text{LUMO}}$ | $\langle R^2 \rangle$    | ZPVE     | $U_0$      | $U$   | $H$   | $G$     | $c_v$  | $\Delta m$ | $\Delta^{*} m$ |
> | ------ | :----: | :----: | :------: | :------: | :----: | ----- | ------ | ------ | ------ | ------ | ---- | ------ | ------ |
> | STL (reproduced)| 0.06 | 0.19 | 64.47 | 50.53 | 0.44 | 4.38 | 57.15 | 64.27 | 53.96 | 50.57 | 0.07 | - | - |
> | STL (Liu et. al.) | 0.07 | 0.18 | 60.58 | 53.92 | 0.50 | 4.54 | 58.84 | 64.24 | 63.85 | 66.22 | 0.07 | - | - |
> | Unitary| 0.10 | 0.30 | 71.17  | 86.13  | 4.94 | 13.75 | 139.40 | 140.04 | 140.64 | 135.98 | 0.12 | 166.2  | 190.2  |
> | UM     | 0.41 | 0.41 | 161.59 | 149.79 | 1.00 | 4.84  | 65.23  | 65.83  | 66.02  | 64.96  | 0.12 | 106.4  | 117.8  |
> | RLW    | 0.10 | 0.33 | 71.02  | 88.66  | 5.77 | 14.29 | 152.90 | 153.74 | 154.08 | 149.42 | 0.13 | 193.5  | 220.7  |
> | DWA    | 0.11 | 0.32 | 73.95  | 92.37  | 4.91 | 13.33 | 144.93 | 145.67 | 145.99 | 141.58 | 0.13 | 172.2  | 196.7  |
> | PCGrad | 0.11 | 0.29 | 77.06  | 88.83  | 3.88 | 8.91  | 117.92 | 118.58 | 118.42 | 115.64 | 0.11 | 125.4  | 145.1  |
> | FAMO   | 0.17 | 0.30 | 97.23  | 98.01  | 1.60 | 4.94  | 73.58  | 73.97  | 73.86  | 72.79  | 0.10 | 63.5   | 74.8   |
> | AutoScale (ours)   |  0.16 | 0.35 | 92.46 |93.32	|2.61	|6.37|	71.71	|72.25	|72.31|	71.01	|0.09	|80.8	|94.2   |
>
> Table notes: All metrics in the table below are interpreted as "lower is better". $\Delta m$ is computed against the STL performances provided by Liu et. al, while $\Delta^{*} m$ denotes that computed against our reproduced STL results.
>
> $\ $
>
> [1] Align-MTL Senushkin, D., Patakin, N., Kuznetsov, A., & Konushin, A. Independent component alignment for multi-task learning. In Proceedings of the IEEE/CVF Conference on Computer Vision and Pattern Recognition. 2023.
>
> [2] QM9: Blum, L. C., & Reymond, J. L. 970 million druglike small molecules for virtual screening in the chemical universe database GDB-13. Journal of the American Chemical Society, 131(25), 8732-8733. 2009.
>
> [3] FAMO: Liu, B., Feng, Y., Stone, P., & Liu, Q. Famo: Fast adaptive multitask optimization. Advances in Neural Information Processing Systems. 2024.
>
> $\ $
>
> **W3 Code:** Our code will be published upon acceptance as mentioned in line 77.
>
> $\ $
>
> **W4 Typo:** We are grateful for pointing out the mistakes and we have updated them in the new manuscript in lines 310, 314.

---

> > ### Author Response · Authors · 2024-11-23
> > **Response to Reviewer CV7C - Q1~Q2**
> >
> > **Q1 Discussion about the metrics:**
> >
> > From our observation, linear scalarization shows correlations with some metrics, such as gradient magnitude similarity and condition number; while some metrics are independent, such as gradient conflict measures (e.g., cosine similarity), loss descending rate, and improvement ratio.
> >
> > For gradient conflict measures like cosine similarity, a prior work [4] gives us insights. It observes that gradient conflicts naturally arise not just in MTL settings, but also in single-task training scenarios, where gradient signals from different samples may conflict. We quote, that empirical evidence shows that *"conflicts arising from gradient alignment between tasks are not exclusive and can even be more pronounced between different samples within a task" [4]. Based on this, we suspect that unless severe gradient conflicts systematically occur during training, the convergence is less likely to be affected. Another interesting finding of Royer et al. [5], is that it is rather common to encounter more gradient conflicts towards the end of training, while the losses of tasks nevertheless steadily decrease. This may suggest that gradient conflict is not a metric that necessarily requires minimization for good model performance.
> >
> > Metrics like loss descending rate and improvement ratio, on the other hand, can be considered as a derivative of the task losses w.r.t. time. Since loss curves are often noisy, these metrics are largely affected by the noise and may not produce meaningful or robust output.
> >
> > We believe that further experimentation with linear scalarization across a broader range of MTO metrics, datasets, and models will deepen our understanding of these relationships.
> > Our work provides an important starting point for exploring the interplay between MTO metrics and linear scalarization weights.
> >
> > $\ $
> >
> > [4] Elich, C., Kirchdorfer, L., Köhler, J. M., & Schott, L. (2023). Challenging Common Assumptions in Multi-task Learning. arXiv preprint arXiv:2311.04698.
> >
> > [5] Royer, A., Blankevoort, T. \& Ehteshami Bejnordi, B. Scalarization for multi-task and multi-domain learning at scale. Advances In Neural Information Processing Systems. 2024.
> >
> > $\ $
> >
> > **Q2 Benefits of *AutoScale* compared with traditional MTO algorithms:**
> >
> > We summarize our advantages below:
> >
> > - **Effectiveness**: *AutoScale* shows superior performance across multiple datasets consistently, compared with various MTO baselines.
> >
> > - **Efficiency**: Unlike gradient manipulating algorithms such as Aligned-MTL, GradNorm, PCGrad, IMTL-G, our *AutoScale* (in its default setting) requires gradient computation only during the exploration phase, which is 20\% of the total training time. The remaining 80\% phase is pure linear scalarization, which is the most efficient. Thus, we are comparably more efficient.
> >
> > - **Flexibility**: According to the formulation of our method, *AutoScale* accepts the use of any heuristics cost function to estimate the optimal linear scalarization weight.
> > It does not have to be limited to what we present in our main paper.
> >
> > $\ $
> >
> > Thank you for your inspiring questions. We hope our responses address your concerns.

---

> > ### Comment · Reviewer_CV7C · 2024-11-24
> > **Thanks for the rebuttal**
> >
> > I read the authors' rebuttal and keep the score.

---

> > > ### Author Response · Authors · 2024-11-27
> > > **Response to Reviewer CV7C**
> > >
> > > Dear Reviewer CV7C,
> > >
> > > Thank you for your review.
> > > We hope our replies have addressed your concerns and questions.
> > > Since the rebuttal period has been extended by a week, please let us know if there is anything further we can clarify or improve.

---

### Official Review · Reviewer_ovJT · 2024-11-02

**Soundness:** 3
**Presentation:** 2
**Contribution:** 2
**Rating:** 5
**Confidence:** 3

**Summary:**

This work studies the relationship between MTO and linear scalarization, which is over debate in the literature. Authors use some metrics to demonstrate the positive relationship. Based on this observation, a new two-phase pipeline for MTL is proposed and demonstrated in multiple datasets.

**Strengths:**

This work is the first to explore relationship between MTO and linear secularization.

Experiments show the benefit of AutoScale in terms of evaluation metrics and running time.

**Weaknesses:**

Notations are not given before their formal definitions, which creates barriers of reading.

Some explanations are not given, for example, why are those cost functions defined in that way?

Results (figure/table) are not clearly explained.

**Questions:**

What are G1, G2, G3, …, B1 in Figure 2?

I feel the whole structure needs to be rewritten. For example, Delta m is in Figure 3 and authors wrote on Line 257 that Figure 3 shows clear correlations. However, Delta m is not defined until Section 5. Key definitions need to be given in the main text.

In Section 4.1, authors propose three cost functions. Are these cost functions related to the literature. Are there any reasonings on choosing these? Why are these functions helpful to the problem? Authors should explain the intuitions.

In Section 4.2, what is the fifth method? Exponential fit? What are the conclusions from Figure 4?

Is Time s/iter the running time? For example, the last column of Table 1. This should be defined clearly.

Authors show the running time, different values of alpha (Figure 6) etc. If these are to demonstrate the generalization or robustness, authors should show them in all datasets.

---

> ### Author Response · Authors · 2024-11-23
> **Response to Reviewer ovJT - W1~W3, Q1~Q3**
>
> Thank you for your comments to help improve our paper. We have incorporated your feedback and updated the manuscript accordingly.
>
> $\ $
>
> **W1 \& W3:** Thanks for pointing out. We have checked notations, figures, and tables in the paper to make sure they are explained.
> We have updated them in the revised paper, marked in blue, like in lines 116, 184.
> Let us know if there are unclear parts.
>
> $\ $
>
> **W2:** Please see our response to Q3 below.
>
> $\ $
>
> **Q1 Figure legends G1, G2, G3, …, B1:**
>
> To avoid ambiguity, we update the figure legends, replacing "good"/"bad" runs "G1, G2, G3, M, B3, B2, B1" with "R1, R2, R3, R4, R5, R6, R7" (line 185).
> The definitions of these runs are provided in the captions of Figure 2 (line 184).
> They correspond to seven training runs with distinct linear scalarization weight sets on the CityScapes dataset, ranked by their $\Delta m$ performance from highest to lowest: $R1 > R2 > R3 > R4 > R4 > R5 > R6$.
>
> Figure 2 provides straightforward insights: certain MTO metrics exhibit distinct behaviors under different linear scalarization sets, while some metrics show no correlation with performance at all.
>
> $\ $
>
> **Q2:**
>
> **Overall Structure**:
> The structure of our paper reflects the logical and chronological progression of our research.
> Our findings - one of our core contributions - are based on extensive empirical experiments and serve as the foundation for the proposed *AutoScale* pipeline.
> Therefore, these findings are introduced first in Section 3 (MTL Metrics in Linear Scalarization).
> Following this, we present our methodology and framework that build upon these findings in Section 4 (Method), with framework performance results detailed in Section 5 (Experiment).
>
> **${\Delta m}$ in Figure 3 and its definition:**
> Since our findings are based on empirical studies, $\Delta m$ - a commonly used performance metric in the multi-task learning field -  appears as shown in Figure 3.
> It reflects the performance of different linear scalarization weight sets.
> We briefly explain the role of ${\Delta m}$ in the figure caption and refer readers to Section 5. Evaluation Metrics for a detailed explanation in the original manuscript.
> To further enhance clarity, we have added a brief sentence in the Figure 3 caption to provide immediate context in line 225.
>
> We hope this revision and our replies resolve your questions.
> Please let us know if further clarification is needed, and we will do our best to address your comments.
>
> $\ $
>
> **W2 \& Q3 Cost function:**
>
> - **Intuition: Why might these cost functions help?**
>
>     Our intuition stems from the observed strong relationship between linear scalarization performance and key MTO metrics, as shown in Figure 3. Specifically, high performance correlates with:
>     1. High gradient magnitude similarity,
>     2. Low condition number, and
>     3. Balanced loss scales.
>
>     We hypothesize the reverse also holds: if we can find a set of weights, that optimize these metrics, such as maximizing gradient magnitude similarity, will we get high performance?
>     This hypothesis drives the design of our cost functions, which are formulated to optimize these three key MTO metrics.
>     The rationale for this design is also explained at the beginning of Section 4. Method (line 272).
>
> - **Why do we choose these three metrics?**
>
>     We have analyzed many metrics as shown in Figure 2 and Appendix B.1, but selected these three for the following reasons:
>     1. These metrics show clear correlations with performance, making them strong candidates for optimization, as shown in Figure 3.
>     2. While some other metrics also show correlation, these three are particularly relevant because they represent measure distinct training challenges:
>         - gradient magnitude similarity: gradient dominance,
>         - condition number: training instability,
>         - loss scale: imbalanced loss.
>
>         These challenges are well-summarized in Section 2.2 (Related work: Multi-Task Optimization). We want to tackle these problems in multi-task optimization.
>
> - **Are they related to the literature?**
>
>     Yes, these three metrics are closely related to prior works, as summarized in Section 3.1 (MTL Metrics Summary).
>     1. Gradient magnitude similarity: Proposed by Yu et al. in "Gradient Surgery for Multi-Task Learning" (NeurIPS 2020) [1]. (line 200)
>     2. Condition number: Introduced in AlignMTL by Senushkin et al. (CVPR 2023) [2]. (line 245)
>     3. Relative loss scale: A simple concept widely used in MTO studies.
>
>     In Section 4.1 (Cost Function), we extend these metrics into cost functions using simple linear matrices, designed to optimize equal gradient magnitude, low condition number, and equal loss scale, respectively.

---

> ### Author Response · Authors · 2024-11-23
> **Response to Reviewer ovJT - Cont'd Q4-Q6**
>
> [1] Yu T, Kumar S, Gupta A, et al. Gradient surgery for multi-task learning[J]. Advances in Neural Information Processing Systems, 2020, 33: 5824-5836.
>
> [2] Senushkin D, Patakin N, Kuznetsov A, et al. Independent component alignment for multi-task learning[C]//Proceedings of the IEEE/CVF Conference on Computer Vision and Pattern Recognition. 2023: 20083-20093.
>
> $\ $
>
> **Q4 Typo and Figure 4:**
>
> Thanks for pointing this out. This is a typo. "five" should be "four". We have fixed the typo and changed the text to make the connection between the paragraph and the figure clearer in the updated version in line 346.
>
> Figure 4 illustrates four simple methods for determining fixed weights, and helps readers understand the process.
> During the exploration phase (the first 20\% of training), we collect statistics such as task gradients and losses to calculate a set of locally optimized weights (indicated by the purple line).
> These weights then serve as inputs to determine the fixed weights for the remaining training period.
> The four fixed weights are represented by the horizon lines in the figure, corresponding to the four methods.
> The performances of each choice are shown in Section 5.1 Ablation Study. Ablation on f({w}) in line 506.
>
> $\ $
>
> **Q5 s/iter:**
>
> Thanks for pointing this out. Time metrics - s/iter - means second per training iteration.
> We have added the explanation of s/iter in the caption of Table 1 in line 382.
>
> $\ $
>
> **Q6 Experiments:**
>
> We appreciate the reviewer’s suggestion. In addition to the extensive experiments already performed across three datasets (Tables 1, 2, 4, 5), we have added the metric behavior across these datasets (refer to our reply to reviewer GjG7), and added further runtime within the limited rebuttal period.
>
> - **Runtime:**
>     We have added the running time for NYUv2 and Cityscapes datasets besides Nuscene, in **Appendix D.1 Runtime** of the revised paper in line 1022, which we also attach below. Note that the iteration time of our method, $T_{\text{AutoScale}}$, can be estimated by $T_{\text{AutoScale}} = \alpha T_{\text{MTO}} + (1 - \alpha )T_{\text{LS}}$, where $\alpha$ is the exploration ratio introduced in line 284, $T_{\text{MTO}}$ refers to the iteration time of the MTO used for the exploration phase, and $T_{\text{LS}}$ is that of linear scalarization. In our default setting, $\alpha = 0.2$ and we use IMTL-G in the exploration phase. Generally, *AutoScale* is more efficient than gradient manipulating MTO algorithms such as GradNorm, MGDA, IMTL-G, and Aligned-MTL, which require gradient computation throughout the entire training process.
>
>     | Method | Nuscene |  | CityScapes |  | NYUv2 |  |
>     |---|---|---|---|---|---|---|
>     |  | Iter. Time  (s) | Relative  Time | Iter. Time  (s) | Relative  Time | Iter. Time  (s) | Relative  Time |
>     | Linear Scalization | 0.453 | 1.00 | 0.195 | 1.00 | 0.298 | 1.00 |
>     | Various MTOs |  |  |  |  |  |  |
>     | UM | 0.455 | 1.01 | 0.199 | 1.02 | 0.367 | 1.23 |
>     | Gradnorm | 1.130 | 2.50 | 0.572 | 2.93 | 0.790 | 2.65 |
>     | MGDA | 1.157 | 2.56 | 0.446 | 2.29 | 0.747 | 2.51 |
>     | PCGrad | 1.170 | 2.59 | 0.416 | 2.13 | 0.765 | 2.57 |
>     | IMTL | 1.158 | 2.56 | 0.422 | 2.16 | 0.829 | 2.78 |
>     | RLW | 0.455 | 1.01 | 0.190 | 0.97 | 0.287 | 0.96 |
>     | Aligned-MTL | 1.213 | 2.68 | 0.430 | 2.21 | 4.144 | 13.91 |
>     | FAMO | 0.457 | 1.01 | 0.198 | 1.02 | 0.290 | 0.97 |
>     | Auto Scale (Ours) | 0.591 | 1.31 | 0.261 | 1.34 | 0.431 | 1.45 |
>
> - **Ablation studies:**
>
>     We acknowledge the importance of ablation studies on all datasets but regret that this is not fully feasible due to resource and time constraints.
>     For example, training on the nuScenes dataset with UniTR (the model we use), requires over 20 hours on 8 GPUs for a single run, making it challenging to perform a comprehensive ablation analysis.
>
>     Despite these limitations, as presented in the main paper, we have conducted the ablation analyses on the cost function and weight predictor f(w) across all three datasets (presented in Tables 4 and 5 of the main paper).
>
> $\ $
>
> We believe our work offers valuable contributions to the community, as highlighted in our general reply. We have carefully addressed all the concerns you raised and made improvements accordingly. Please feel free to reach out if you have any additional questions or require further clarification.

---

> > ### Comment · Reviewer_ovJT · 2024-11-23
> >
> > I thank authors for their responses with the new experiments. Clear writing and definition can make the paper easier to follow. I will raise my score.

---

> ### Author Response · Authors · 2024-11-24
> **Response to Reviewer ovJT**
>
> Dear Revier ovJT,
>
> We are grateful for your response and that you find the clarity problems to now be addressed.  We would appreciate if you could point out the remaining weaknesses that you find in the paper that cause you to recommend marginally below acceptance. Thank you!

---

> > ### Author Response · Authors · 2024-11-27
> > **A reminder for Reviewer ovJT**
> >
> > Dear Reviewer ovJT,
> >
> > We kindly follow up on our message from three days ago regarding any remaining weaknesses you see in the paper leading to your recommendation. Your feedback would be greatly appreciated. Thank you!

---

### Official Review · Reviewer_GjG7 · 2024-11-04

**Soundness:** 3
**Presentation:** 3
**Contribution:** 3
**Rating:** 6
**Confidence:** 3

**Summary:**

This paper explores the connection between multi-task optimization (MTO) and linear scalarization, proposing the use of multi-task learning metrics such as gradient magnitude similarity and condition number to guide the determination of weights in linear scalarization. The proposed approach, named AutoScale, includes detailed discussions on the initial multi-task learning phase and various strategies for optimizing the weights, which gives insights into its implementation.

**Strengths:**

Multi-task optimization and learning is essential in machine learning. The approach of using multi-task learning metrics to determine the weights in the linear scalarization seems novel. The proposed method is computationally more feasible than an exhaustive grid search of weights.

**Weaknesses:**

1. While using MTL metrics to estimate the optimal weight appears reasonable based on the results shown in Figure 2, it is unclear whether these findings would hold consistently across other datasets. Additionally, the evaluation metric—a single average of the accuracy of multiple tasks—may not fully capture the nuanced definition of "good" and "bad" results, especially in a multi-task learning context. More careful discussions are needed to justify the use of MLT metrics.
2. The paper could benefit from more comparisons with linear scalarization approaches that directly use predictive performance metrics as criteria for determining weights.
3. Beyond the unitary and grid-searched weights discussed, the authors might consider exploring other weight determination methods, such as those surveyed by Royer et al. (2024), to provide a broader context and benchmark for AutoScale.

**Questions:**

In Table 1, the iteration time is reported. Could the authors also provide the total computational time? Additionally, what is the computational time for each method shown in Table 2?

---

> ### Author Response · Authors · 2024-11-23
> **Response to Reviewer GjG7 -  W1**
>
> Thanks for the valuable comments! We have addressed your concerns and questions individually below.
>
>
> **W1 (a) Findings' consistency across datasets:**
>
> We have added an analysis of MTO metrics' behaviors on two additional datasets - NYUv2 and Nuscene, in the revised manuscript, under **Appendix B. More Metrics Visualization in Linear Scalarization Across Various Datasets** in line 778.
>
> From Figures 10, 11, 12, and 13, we observe that our findings generally hold across the three datasets:
>
> - Regarding the 3 core MTO metrics (shown in Figure 3) - gradient magnitude similarity, condition number, and balanced loss scale: they keep serving as a relatively good indicator for performance.
> Gradient magnitude similarity and condition number consistently show a strong correlation with the performance of linear scalarization in both datasets as shown in Figures 10, 11, 12, 13, as similar patterns in Figures 2 and 3 in CityScapes.
> For the balanced loss (measured by the standard deviation of relative loss), the correlation is less robust, though it can be observed that worse performance (higher $\Delta m$) is always along with a large standard deviation of loss among tasks.
> - Metrics that are not correlated with the performance are consistently observed including loss variance, and cosine similarity (as linear scalarization does not change per-task gradient direction).
> - Other metrics show correlation sometimes but are not stable, which might not be good indicators for performance, including projected magnitude, and inverse learning rate.
>
> While we attempt to experiment over multiple benchmarks, we cannot guarantee the generality of the findings across all models and datasets.
> Nevertheless, we believe that exploring the correlation between MTO metrics and linear scalarization, and our findings across the tested datasets, are valuable and interesting to the community, as one of our core contributions.
>
>
>
> **W1 (b) Metrics:**
>
> We agree that defining a suitable evaluation metric for measuring multi-task learning performance is challenging, as in many other research areas.
>
> - **More metrics beyond ${\Delta m}:$**
> We rank the performance using ${\Delta m}$ in Figures 2 and 3 because it is one of the most widely used metrics in MTL.
> ${\Delta m}$ measures the average performance drop compared to single-task learning across all tasks.
> Indeed, it alone might not fully capture all performance nuances.
> Therefore, we also include the performance of seven runs used in Figure 2 using additional metrics, including Mean Rate (MR) and our proposed metric ${\Delta m_{pos}}$, as the same way reported in Table 1 \& 2.
> In table below, the performance rankings are largely consistent across all three metrics, with only slight variations in MR for run R4 (revised) / M (old).
>
> - **Introducing New metrics ${\Delta m_{pos}}$:**
> Exactly, to address limitations of existing metrics including ${\Delta m}$ and mean rank, we emphasize one of our contributions: we introduce a new metric, ${\Delta m_{pos}}$, detailed in Section 5. Experiment. Evaluation Metrics in line 416.
> Unlike $\Delta m$, which measures average performance changes (both positive and negative) compared to single-task learning, ${\Delta m_{pos}}$ focuses exclusively on total performance drops, disregarding any improvements.
> This provides a valuable perspective, especially for cases where minimizing overall performance drops is prioritized over allowing sacrifices in some tasks to boost others, as we describe in line 422 of the manuscript.
>
> | Method | $\Delta m\downarrow$ | MR$\downarrow$ | $\Delta m_{pos}\downarrow$ | Semantic Seg. mIoU [\%] | Instance Seg. L1 [pxl] | Disparity MSE |
> |---|:---:|:---:|:---:|:---:|:---:|:---:|
> | STL Baseline | - | - | - | 66.73 | 10.55 | 0.330 |
> | Different linear scale. |  |  |  |  |  |  |
> | G1 (old) $\rightarrow$ R1 (revised) | -1.42 | 2.67 | 0.69 | 66.27 | 10.36 | 0.320 |
> | G2 (old) $\rightarrow$ R2 (revised) | -0.74 | 3.00 | 2.61 | 64.99 | 10.08 | 0.329 |
> | G3 (old) $\rightarrow$ R3 (revised) | 0.10 | 4.00 | 1.62 | 65.65 | 10.42 | 0.330 |
> | M (old) $\rightarrow$ R4 (revised) | 2.98 | 3.00 | 15.30 | 60.04 | 9.88 | 0.347 |
> | B3 (old) $\rightarrow$ R5 (revised) | 7.43 | 5.00 | 26.29 | 56.63 | 10.13 | 0.367 |
> | B2 (old) $\rightarrow$ R6 (revised) | 9.59 | 5.00 | 34.53 | 54.10 | 9.94 | 0.382 |
> | B1 (old) $\rightarrow$ R7 (revised) | 10.62 | 5.33 | 37.47 | 54.16 | 9.96 | 0.392 |

---

> ### Author Response · Authors · 2024-11-23
> **Response to Reviewer GjG7 - W2, W3, Q1,**
>
> **W2 \& W3 Comparision with performance-based weight search methods:**
>
> We add a comparison of performance-based weight search methods with our proposed *AutoScale* on the CityScapes dataset.
> Specifically, we evaluate:
>
> - Bayesian optimization: a probabilistic model-based method for global optimization of black-box functions. Based on the performance of previous N trials, it constructs a Gaussian process distribution and explores weight combinations that are most likely to improve performance.
> - Population-based training (PBT) surveyed by Royer et al. (2024): N trials trained in parallel starting with different initialized weights set. Every $E_{ready}$ epochs, the worst-performing Q\% models, based on performance rankings, are stopped. Their model parameters and task weights are replaced by one of the best-performing Q\% models, with adding some perturbations to the weights to continuously find the best weights.
>
>     |  | # of Trials | Total GPU$\times$hours | $\Delta m \downarrow$ |
>     |---|---|---|---|
>     | AutoScale (Ours) | 1 | 8 | 0.10 |
>     | Bayes Search | 4 | 29 | 0.29 |
>     |  | 8 | 58 | \-0.06 |
>     | PBT Search | 2 | 23 | 0.13 |
>     |  | 4 | 40 | 0.31 |
>     |  | 8 | 74 | 0.14 |
>
> Due to the time and resource limits, we test on a small number of trials: [4,8] trials for Bayesian search; and [2,4,8] for PBT search, with performance check every $E_{ready}=20$ epochs out of a total 100 epochs, and Q\% is set to [0.5, 0.25, 0.25] respectively (i.e. half or a quarter worse models are replaced by better-performing ones with weight perturbation).
>
> Performance-based weight search methods rely on predictions from previous N trials to refine weight selection.
> As shown in the table, their performance is quite random and less robust when N is small due to the randomness of N initialized weights.
> While these methods achieve more robust and reliable results when N is large,
> the computational cost (GPU$\times$hours) increases significantly as N grows.
>
> In contrast, *AutoScale* is more efficient as it determines the weights in a single trial while maintaining competitive performance, based on cost function optimization of key MTO metrics during the training.
> It shows the scalability of our method (and also most MTO algorithms), compared to general performance-based methods, particularly with large models and datasets.
>
> $\ $
>
> **Q1 Runtime:**
>
> We report the total computational time for each method on the Nuscenes dataset in Table 1 below. All experiments are done with 8 $\times$ A100 GPUs, detailed in Appendix D. Experiments Details.
>
> | Method | Training Time (h) | Iter. Time (s) |
> |---|---|---|
> | Unitary | 19.53 | 0.453 |
> | Aligned-MTL | 30.98 | 1.213 |
> | FAMO | 19.60 | 0.457 |
> | IMTL-G | 30.08 | 1.158 |
> | MGDA | 29.97 | 1.157 |
> | PCGrad | 30.38 | 1.170 |
> | RLW | 19.73 | 0.455 |
> | UM | 19.89 | 0.455 |
> | Gradnorm | 30.15 | 1.130 |
> | AutoScale (ours) | 22.78 | 0.591 |
>
> We have also added the runtime for each method across three datasets, in **Appendix D.1 Runtime** of the revised paper in line 1022, which we also attach below. Note that the iteration time of our method, $T_{\text{AutoScale}}$, can be estimated by $T_{\text{AutoScale}} = \alpha T_{\text{MTO}} + (1 - \alpha )T_{\text{LS}}$, where $\alpha$ is the exploration ratio introduced in line 284, $T_{\text{MTO}}$ refers to the iteration time of the MTO used for the exploration phase, and $T_{\text{LS}}$ is that of linear scalarization. In our default setting, $\alpha = 0.2$ and we use IMTL-G in the exploration phase. Generally, *AutoScale* is more efficient than gradient manipulating MTO algorithms such as GradNorm, MGDA, IMTL-G, and Aligned-MTL, which require gradient computation throughout the entire training process.
>
> | Method | Nuscene |  | CityScapes |  | NYUv2 |  |
> |---|---|---|---|---|---|---|
> |  | Iter. Time  (s) | Relative  Time | Iter. Time  (s) | Relative  Time | Iter. Time  (s) | Relative  Time |
> | Linear Scalarization | 0.453 | 1.00 | 0.195 | 1.00 | 0.298 | 1.00 |
> | Various MTOs |  |  |  |  |  |  |
> | UM | 0.455 | 1.01 | 0.199 | 1.02 | 0.367 | 1.23 |
> | Gradnorm | 1.130 | 2.50 | 0.572 | 2.93 | 0.790 | 2.65 |
> | MGDA | 1.157 | 2.56 | 0.446 | 2.29 | 0.747 | 2.51 |
> | PCGrad | 1.170 | 2.59 | 0.416 | 2.13 | 0.765 | 2.57 |
> | IMTL | 1.158 | 2.56 | 0.422 | 2.16 | 0.829 | 2.78 |
> | RLW | 0.455 | 1.01 | 0.190 | 0.97 | 0.287 | 0.96 |
> | Aligned-MTL | 1.213 | 2.68 | 0.430 | 2.21 | 4.144 | 13.91 |
> | FAMO | 0.457 | 1.01 | 0.198 | 1.02 | 0.290 | 0.97 |
> | AutoScale (ours) | 0.591 | 1.31 | 0.261 | 1.34 | 0.431 | 1.45 |
>
> $\ $
>
> Thank you for your insightful questions, which have helped us clarify and strengthen our work through additional experiments and detailed responses.

---

> > ### Comment · Reviewer_GjG7 · 2024-11-25
> >
> > Thank you very much for your efforts in addressing my questions and comments. My score remains the same!

---

> ### Author Response · Authors · 2024-11-27
> **Response to Reviewer GjG7**
>
> Dear Reviewer GjG7,
>
> Thank you for acknowledging our efforts in addressing your questions and comments.
> We hope our responses have addressed your concerns.
> Since the rebuttal period has been extended by a week, please let us know if there is anything further we can clarify or improve.

---

### Author Response · Authors · 2024-11-24
**General reply**

We sincerely thank all the reviewers for their insightful and detailed feedback.
We have addressed each reviewer's comments individually, and would like to use this global response to highlight our contributions and the experiments that we have newly added.

$\ $

**Strength and Contribution**:

We appreciate that the reviewers acknowledge the strengths of our study:

- **Novelty:** Our work is the first to explore the relationship between MTO and linear scalarization.
Motivated by the correlations, we propose a novel perspective to use multi-task learning metrics to determine the weights in linear scalarization, as pointed out by Reviewer **ovJT, GjG7**.
- **Effectiveness and Efficiency:** *AutoScale* consistently shows good performance across various datasets via comprehensive experiments, with lower computational costs compared with gradient manipulating MTO algorithms or weight search methods, as pointed out by Reviewer **CV7C, GjG7**.
- **Clarity:** The well-organized paper and clear idea are appreciated by Reviewer **CV7C**.


In addition to the above points, we emphasize that beyond its effectiveness and efficiency, *AutoScale* is also highly flexible.
By design, *AutoScale* can incorporate any heuristic cost function to estimate the optimal linear scalarization weight.
It is not limited to the three MTO metrics optimizations presented in our main paper.
We hope our work inspires further exploration into leveraging the core principles of various MTO algorithms to assist in weight search for linear scalarization.

$\ $

**Updates to the Reviewers' Feedback:**

In response to the valuable suggestions from reviewers, we have added the following new analyses and experiments.

- We expand our analysis of the correlation between various MTO metrics and performance to include the NYUv2 and nuScenes datasets, in addition to the CityScapes dataset presented in the manuscript. (Refer to the comments from Reviewer **GjG7**.)
- We add performance comparisons on QM9 dataset, which contains 11 tasks. This demonstrates our method's capability to scale up to a higher number of tasks. (refer to the comments from Review **CV7C**)
- We further include a comparison with performance-based weight search methods, namely Bayesian Optimization (BO) and Population-Based Training (PBT). (refer to the comments from Review **GjG7**)
-  We add the runtime analysis of our method across all datasets compared with all baselines. (refer to the comments from Review **GjG7, ovJT**)

$\ $

Finally, we would be happy to address any further questions during the discussion period.

---

### Meta-Review · Area_Chair_3TF6 · 2024-12-13

**Metareview:**

The paper studies the relations between linear scalarization and multi-task optimization algorithms. Extensive experiments are also provided in the paper. But reviewers find that the technical contributions are very limited. The problem studied in this paper is not very important and may not attractive to more researchers. None of reviewers shows the strong support for this paper.

**Additional Comments On Reviewer Discussion:**

Reviewes still thinks this paper is below the acceptance bar.

---

### Decision · Program_Chairs · 2025-01-22

Reject